# Understanding Noise-Augmented Training for Randomized Smoothing

**Ambar Pal**  AMBAR@JHU.EDU
*Department of Computer Science &*
*Mathematical Institute of Data Science*
*Johns Hopkins University*
*Baltimore, MD 21218, USA*

**Jeremias Sulam**  JSULAM1@JHU.EDU
*Department of Biomedical Engineering &*
*Mathematical Institute of Data Science*
*Johns Hopkins University*
*Baltimore, MD 21218, USA*

**Reviewed on OpenReview:** *https://openreview.net/forum?id=fvyh6mDWFr*

## Abstract

Randomized smoothing is a technique for providing provable robustness guarantees against adversarial attacks while making minimal assumptions about a classifier. This method relies on taking a majority vote of any base classifier over multiple noise-perturbed inputs to obtain a smoothed classifier, and it remains the tool of choice to certify deep and complex neural network models. Nonetheless, non-trivial performance of such smoothed classifier crucially depends on the base model being trained on noise-augmented data, i.e., on a smoothed input distribution. While widely adopted in practice, it is still unclear how this noisy training of the base classifier precisely affects the risk of the robust smoothed classifier, leading to heuristics and tricks that are poorly understood. In this work we analyze these trade-offs theoretically in a binary classification setting, proving that these common observations are not universal. We show that, without making stronger distributional assumptions, no benefit can be expected from predictors trained with noise-augmentation, and we further characterize distributions where such benefit is obtained. Our analysis has direct implications to the practical deployment of randomized smoothing, and we illustrate some of these via experiments on CIFAR-10 and MNIST, as well as on synthetic datasets.

## 1 Introduction

Machine learning classifiers are known to be vulnerable to adversarial attacks, wherein a small, human-imperceptible additive perturbation to the input is able to produce a change in the predicted class (Szegedy et al., 2013). Because of clear security implications (Kurakin et al., 2016), this phenomenon has sparked increasing amounts of work dedicated to devising defense strategies (Metzen et al., 2017; Gu and Rigazio, 2014; Madry et al., 2017) and correspondingly more sophisticated attacks (Carlini and Wagner, 2017; Athalye et al., 2018; Tramer et al., 2020), with each group trying to triumph over the other in an arms-race of sorts. As a result, an increasing number of works have begun providing theoretical guarantees and understanding of adversarial attacks and defenses, studying learning theoretic questions (Shafahi et al., 2018; Cullina et al., 2018; Bubeck et al., 2018; Tsipras et al., 2018), understanding the sample complexity of robust learning (Schmidt et al., 2018; Yin et al., 2018; Tu et al., 2019; Awasthi et al., 2019), characterizing the optimality of attacks and defenses (Pal and Vidal, 2020), analyzing the implications of robust representations (Awasthi et al., 2020; Sulam et al., 2020; Muthukumar and Sulam, 2022), and more.

One of the central objects of study in this setting are robustness certificates, which provide provable guarantees on the prediction of models as long as the input is not perturbed beyond a specific contamination level (Cohen et al., 2019; Raghunathan et al., 2018; Yang et al., 2020). In this vein, randomized smoothing (Lecuyer et al., 2019; Cohen et al., 2019) employs a *base* classifier, typically vulnerable to adversarial attacks, and derives from it a *smoothed* version by taking a majority vote of the base classifier's outputs on several (stochastic) noise-perturbed versions of the input. While making minimal assumptions about the base model, the resulting smoothed predictor is provably robust to input perturbations of bounded $\ell_p$ norm (Yang et al., 2020). Because this is applicable to *any* given predictor, randomized smoothing has arguably become the most practical certified defense technique against adversarial examples for deep learning models, which are often too complex be analyzed and certify otherwise.

Nonetheless, the accuracy of the resulting smoothed model differs from that of its base classifier. It has been empirically observed that smoothing a base classifier *out of the box*, i.e., without any modifications to the network weights or structure, does not provide good performance, for example in terms of certified accuracy (Cohen et al., 2019; Gao et al., 2019). Tricks like noise-augmented training for the base classifier need to be employed in order to obtain meaningful results and defense certificates. This phenomenon, while pervasive in practice, does not have a good theoretical understanding. There are several open questions surrounding the relationship between the base classifier and its smoothed version: Why is noise-augmented training useful when training the base classifier? How does the performance of the smoothed classifier depend on the distribution of the training noise? Is noise perturbed training always needed for all data-distributions, and if not, could we determine this before modifying the training procedure? In this paper, we take a step towards answering some of these questions for a general binary classification task in $\mathbb{R}^d$.

In summary, our contributions are as follows:

1. For a general bounded data-distribution, we derive an upper bound on the benign risk obtained from smoothing a classifier that has been trained with noise augmentation. Surprisingly, this bound suggests that noise-augmented training can in fact be harmful to the benign risk of the final smoothed classifier, which is contrary to the observed phenomenon in practical applications.

2. We then characterize a family of data distributions for which the above bound is tight, and for which training on noise-augmented data is only detrimental. These distributions are characterized by a notion of large *interference distance*, which we formalize.

3. We show that this notion of interference distance captures some of the trade-offs in randomized smoothing, and we prove that there exist a family of distributions where benefits can be obtained from noise-augmented training – as observed in practice.

4. Our empirical experiments suggest, firstly, that real data distributions lie in the low-interference distance regime, and hence noise-augmented training is beneficial to randomized smoothing. Secondly, contrary to practice, our proofs suggest that the parameters of the noise-distribution for noise-augmented training and that of randomized smoothing need not be the same, and that allowing for different values of these parameters lead to improved results. Our experiments on MNIST and CIFAR-10 confirm this theoretical intuition.

The rest of the paper is organized as follows. In Section 2 we describe preliminaries and set up our notation and framework. In Section 3 we introduce our main results informally, obtaining conditions that characterize data distributions where noise-augmented training provides an advantage, and those where it does not. Section 4 presents our main results in greater technical detail. Lastly, we conduct our synthetic and real data experiments in Section 5, and conclude in Section 7 highlighting our answers to the questions posed above.

## 2   Preliminaries and Setup

We consider a binary classification [1] task on data $X \in \mathcal{X} \subset \mathbb{R}^d$ with labels $Y \in \mathcal{Y} = \{0, 1\}$. The random variable $X$ follows a data distribution over the space $\mathcal{X}$ with PDF $p_X$. In turn, the label or response variable

---

[1]We comment on extensions to multiple classes later in Section 4.

$Y$ follows a distribution $p_Y$ over its corresponding space. We assume that $\mathcal{X}$ is bounded (we take diameter 1 for simplicity). A supervised learning task is defined by the joint distribution $p_{X,Y}$, and the goal is to obtain a predictor $h\colon \mathcal{X} \to [0,1]$ so that $h$ is a good approximation for the conditional expectation $\mathbb{E}[Y|X = x]$. These distributions are unknown, however, so the learning problem consists of finding such a predictor from a set of samples, typically identically and independently distributed from $p_{X,Y}$.

In this work we will not dwell much on the learning problem, and instead we will assume we are given access to a function $h\colon \mathcal{X} \to [0,1]$ which is a good approximation [2] for the conditional $p_{Y|X}$. The *base* classifier is given by the composition of $h$ with a discretizing decision mapping, $\psi\colon [0,1] \to \mathcal{Y}$. In our setting, $\psi(z) = \mathbb{1}[z \geq 0.5]$, where $\mathbb{1}[\cdot]$ denotes the indicator function. Note that if $h$ is the real conditional distribution of the label for a given input, this coincides[3] with the Bayes classifier for this problem.

Adversarial examples are "small" additive perturbations designed for an input sample such that the predicted label at the perturbed input is incorrect. Typically these perturbations are constrained to be in some $\ell_q$ norm ball, i.e., $\|\delta\|_q \leq \epsilon$, so as to be imperceptible to humans. It has been extensively shown that models that achieve excellent accuracy in normal settings can misclassify samples perturbed even with very small values of $\epsilon$ (Goodfellow et al., 2014). The goal of certified robustness methods is to guarantee that the output of a certain model at a given input[4], $\psi(h)(x)$, will not change when contaminated by perturbations in a $\ell_q$-ball with radius of at most $\epsilon^*$. That is, certifying that $\psi(h)(x) = \psi(h)(x + \delta)$ for every $\delta : \|\delta\|_q \leq \epsilon^*$.

Randomized smoothing (Cohen et al., 2019) achieves this by "smoothing" the classifier $\psi(h)$ with an isotropic distribution. More precisely, the smoothed classifier $\mathrm{Smooth}(\psi(h))$ is constructed from $\psi(h)$ by

$$\mathrm{Smooth}_{p_V}(\psi(h))(x) \overset{\text{def}}{=} \arg\max_{c \in \mathcal{Y}} \mathbb{P}[\psi(h)(x + V) = c], \tag{1}$$

where $V \sim p_V$. The choice of the distribution $p_V$ centrally depends on the norm constraint of the adversarial perturbation. Randomized smoothing was first introduced as a certification method against $\ell_2$-bounded perturbations, for which $p_V = \mathcal{N}(0, \beta^2 I_d)$. However, this has been extended to other norms in a series of recent works (Lecuyer et al., 2019; Li et al., 2019; Yang et al., 2020), yielding correspondingly different distributions $p_V$. In particular, for the binary case with $\ell_2$-bounded perturbations, randomized smoothing measures the probability of the predicted class (say 1) after smoothing as $s = \mathbb{P}(h(x + V) = 1)$. Then, it provides a certified radius $\epsilon^* = \beta \Phi^{-1}(s)$ that depends on this probability $s \in (0.5, 1]$ as well as the variance of the smoothing distribution $\beta^2$, where $\Phi^{-1}$ is the inverse of the standard Gaussian CDF. Before continuing, it will be useful for our discussions to employ the following equivalent form of randomized smoothing.

**Proposition 2.1.** *When $p_V$ is symmetric, and for the binary classification task defined above, the smoothed classifier is given by $\mathrm{Smooth}_{p_V}(\psi(h))(x) = \psi((\psi(h) * p_V))(x)$, where $*$ denotes the convolution operation.*

*Proof.* Consider the random variable $V \sim p_V$ and simplify $\mathrm{Smooth}_{p_V}(\psi(h))(x)$ as,

$$\mathrm{Smooth}_{p_V}(\psi(h))(x) = \arg\max_{c \in \{0,1\}} \mathbb{P}[\psi(h)(x + V) = c]$$

$$= \psi(\mathbb{P}[\psi(h)(x + V) = 1]) = \psi\left(\int \mathbb{1}[\psi(h)(x + v) = 1] p_V(v) dv\right)$$

$$= \psi\left(\int \psi(h)(x + v) p_V(v) dv\right) = \psi(\psi(h) * p_V)(x). \qquad \square$$

In this manuscript, we will work with single-parameter smoothing distributions $p_\beta$. Accordingly, we will simplify our notation to denote the smoothed classifier as $\mathrm{Smooth}_\beta(\psi(h))(x)$.

We consider the setting where a certain minimum level of robustness is required at deployment, say $\epsilon^*$. In other words, we want our certified radius to be at least $\epsilon^*$. From the previous discussion, this implies that

---

[2]Our results are developed for $h = E[Y|X]$, but we show that most of them can be extended to the approximate case in Appendix G.

[3]Up to the deterministic nature of $\psi(z)$ whenever $z = 0.5$.

[4]For simplicity of notation in our analysis, we use $\psi(h)(x)$ to denote the composition $\psi \circ h$ at $x$.

$\beta\Phi^{-1}(s) \geq \epsilon^*$, implying that we need a minimum strength for randomized smoothing, say $\beta^* \overset{\text{def}}{=} (1/\Phi^{-1}(s))\epsilon^*$. Then, given that one will be required to employ $\text{Smooth}_{\beta^*}(\psi(h))(x)$, how should $h$ be obtained?

To make the above question precise, we recall the definition of the risk of a classifier $f$ under the data-distribution $p_{X,Y}$ as $R(f) = \mathbb{P}(f(X) \neq Y)$. When $f = \text{Smooth}_{\beta^*}(\psi(h))$, this becomes

$$R(\text{Smooth}_{\beta^*}(\psi(h))) = \mathbb{P}(\text{Smooth}_{\beta^*}(\psi(h))(X) \neq Y).$$

As mentioned in the introduction, obtaining $h$ via natural training typically leads to a certifiable predictor with higher error than that of the base classifier, i.e., $R(\text{Smooth}_{\beta^*}(\psi(h))) > R(\psi(h))$. This is expected, as the classifier $\psi(h)$ was precisely trained to minimize the risk $R(\psi(h))$, whereas its smoothed counterpart $\text{Smooth}_{\beta}(\psi(h))$, was not. Such a phenomenon can also be regarded as an out-of-distribution (OOD) problem, albeit in a very specific setting where the distribution shift is produced by a certification method. Indeed, it has been widely demonstrated in practice that naïvely smoothing any base classifier leads to a significant degradation in benign accuracy, i.e., accuracy on non-adversarially-corrupted samples, and hence to a lower certified-accuracy. Besides the well established empirical evidence reported in the literature (Gao et al., 2020; Cohen et al., 2019), we will also showcase this phenomenon in our experiments.

To alleviate the discrepancy, practitioners have resorted to retraining the base classifier $\psi(h)$ with a *noise-augmented* data distribution instead, denoted here by $p_X^s$. This is achieved by adding noise $V \sim p_V$ to the data variable $X \sim p_X$ to obtain the noise-augmented data $X_s = X + V \sim p_X^s$. Following general intuition, in practice one employs the same smoothing distribution for noise-augmentation as that employed for the certification stage. If we let $p_X^s$ be the PDF of $X_s$, it is well known that $p_X^s$ is the convolution of $p_X$ and $p_V$, i.e. $p_X^s = p_X * p_V$.

Throughout this work, we will make minimal assumptions about the underlying data distribution, the parameterization of the predictor, and the (finite) dimension of the data. However, we will assume that the hypothesis class is rich enough, and the learning algorithm good enough, such that the true conditional expectations are learnt successfully. As we show in Appendix B, this implies that the Bayes classifiers $\psi(h)$ and $\psi(h * p_V)$ are learnt successfully under data distributions $P_X$ and $P_X^s$, respectively. Therefore, a classifier learnt on the noise-augmented data results in $\text{Smooth}_{\beta}(\psi(h * p_V))$, which will be the central object of our study.

While the assumption of learning the Bayes classifiers exactly might seem stringent, much of our results can be adapted to relax this assumption while maintaining our general proof technique, assuming a controlled difference between the Bayes and the obtained classifier. In Section 4 and Appendix G, we discuss this further and extend our results to handle errors in learning. On the one hand, our analyses of the Bayes classifiers are informative because they reflect the best possible predictors that can be learned from data. On the other hand, and importantly, we will illustrate how these assumptions are reasonable and sufficient to depict what is observed in relevant scenarios. Indeed, our theoretical results and simulations on synthetic experiments will resemble the observations in natural image data, presented in Section 5.

As earlier, we will employ noise-augmentation distributions parameterized by a single parameter. Accordingly, we will denote our noise-augmentation distribution as $p_\alpha$ to highlight the only parameter, $\alpha$. For instance, when augmenting data with a Gaussian we have that $p_V = p_\alpha = \mathcal{N}(0, \alpha^2 I)$; and in the case of a uniform distribution supported over a $\ell_2$ ball of radius $\alpha$, one has $p_V = p_\alpha = \text{Unif}(B_2(0, \alpha))$. In this way, the final smoothed classifier is given by $\text{Smooth}_{\beta}(\psi(h * p_\alpha))$. A key goal of our work is to understand the interplay between these two parameters: $\beta$ (controlling the certified radius) and $\alpha$ (controlling the noise-augmentation of the data distribution).

## 3 Main Results

Our central aim in this section is to theoretically characterize the empirically observed difference between the benign risks of the original classifier $\psi(h)$, and that of the smoothed classifier trained on noise-augmented data, $\text{Smooth}_{\beta}(\psi(h * p_\alpha))$. In other words, we want to obtain an estimate of the excess risk

$$\Delta_{\alpha,\beta}(h) = R(\text{Smooth}_{\beta}(\psi(h * p_\alpha))) - R(\psi(h)), \tag{2}$$

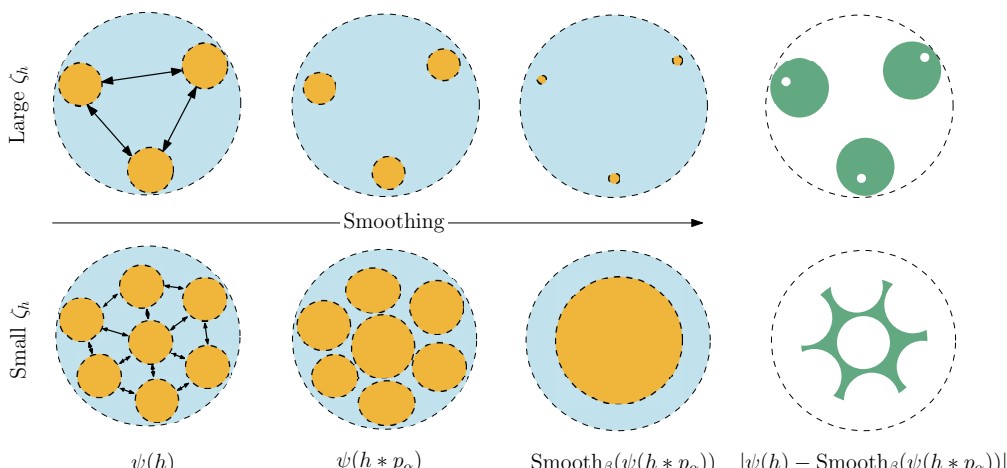

Figure 1: Theorem 3.1 and Theorem 3.3 deal with the cases of large (top row) and small (bottom row) interference distance $\zeta_h$, respectively, where $\zeta_h$ is the average of the pairwise distances between the orange regions, denoted by the black arrows. The text after Theorem 3.3 provides a detailed description of the different regions in the figure.

where the risk measures the probability of error over $p_{X,Y}$ (see Section 1). We will present our result conceptually and slightly informally here, and we will provide a more detailed version of these later in Section 4.

In our first result, we demonstrate that the behaviour observed in practice (namely, that performing noise-augmented training is beneficial[5]) is not universal. We do this by showing that there exists a class of distributions $\mathcal{H}_1$ with a certain minimal *interference distance* property where training the classifier on a noise-augmented distribution before performing the smoothing for certification is in fact detrimental. The interference distance $\zeta_h$ for a classifier $\psi(h)$ can be informally thought of as the average $\ell_2$ distance between any two disjoint input regions classified as 1 by $\psi(h)$ (normalized by the size of the domain). Figure 1 provides a simple illustration, where $\zeta_h$ is the average length of all the black arrows. We defer the formal definition of this property to Section 4, where we will use tighter characterizations by using the minimum and maximum of such lengths.

**Theorem 3.1** (No need for noise-augmentation)**.** *There exists a class of distributions $\mathcal{H}_1$ with large interference distance, such that, for all $h \in \mathcal{H}_1$, training on a smoothed distribution has no benefit; that is $\Delta_{0,\beta}(h) < \Delta_{\alpha,\beta}(h)$ for all $\alpha, \beta > 0$.*

This result is of importance because it sheds light on the kind of behaviour that can be expected while making minimal assumptions of the (conditional) distribution $h$. Indeed, we now present a result that upper bounds $\Delta_{\alpha,\beta}(h)$ as a function of the smoothing parameters, $\alpha, \beta$.

**Theorem 3.2** (Simplified Upper-Bound)**.** *For any $h$ with bounded support, $\Delta_{\alpha,\beta}(h) \leq G_{\alpha,\beta}$, where $G_{\alpha,\beta}$ is a monotonically increasing function of both $\alpha$ and $\beta$.*

We pause to make a few remarks about Theorem 3.2. First, we see that $G_{\alpha,\beta}$ increases with the certification parameter $\beta$, suggesting that the excess risk $\Delta_{\alpha,\beta}$ may increase as we increase $\beta$. This behavior is expected and reflects what is seen in practice. Recall that the inverse of the smoothness parameter of the classifier $\text{Smooth}_\beta(\psi(h * p_\alpha))$ is proportional to the randomized smoothing strength, $\beta$. Hence as $\beta$ increases, the classifier changes more slowly, and more error is incurred in the *spiky* regions of the domain where the true class fluctuates rapidly. This demonstrates the well known accuracy-robustness trade-off: larger $\beta$ leads to better robustness certificates but worse benign performance (Gao et al., 2020; Cohen et al., 2019).

---

[5]Note that we are *not* studying the robust risk of the smoothed classifier. Nevertheless, understanding properties of the benign risk is an essential step towards understanding the robust risk, as the former is a lower bound on the latter.

Secondly, we see that $G_{\alpha,\beta}$ increases monotonically with $\alpha$. This suggests that, for any fixed value of $\beta$, the excess risk $\Delta_{\alpha,\beta}$ is minimized at $\alpha = 0$; i.e, it is better not to perform noise-augmentation during training. This is contrary to the behavior of randomized smoothing in practice: to obtain a smoothed classifier $\mathrm{Smooth}_\beta(\psi(h * p_\alpha))$ with a good benign risk a practitioner typically sets $\alpha = \beta > 0$. One might be tempted to conclude that this seeming contradiction stems from the fact that the bound in Theorem 3.2 is too loose to be informative. Yet, we find that this is not the case. In fact, Theorem 3.2 reflects the behaviour of $\Delta_{\alpha,\beta}(h)$ *for a general h* with bounded support, and this upper bound is tight for functions $h \in \mathcal{H}_1$, i.e., the family of distributions from Theorem 3.1.

If retraining on noise-augmented data is not universally needed, then for what family of distributions is it beneficial? It turns out that our notion of interference distance $\zeta_h$ controls this trade-off, as we now show.

**Theorem 3.3** (Noise-Augmentation helps for distributions with small interference distance)**.** *There exists a class of distributions $\mathcal{H}_2$ with small interference distance, such that, for all $h \in \mathcal{H}_2$, training on noise-augmented data is favorable; i.e., there exists $\alpha, \beta > 0$ such that $\Delta_{\alpha,\beta}(h) < \Delta_{0,\beta}(h)$.*

Before delving into our detailed results in Section 4, we summarize a few key implications of Theorem 3.1 and Theorem 3.3 for randomized smoothing, as illustrated in the example in Figure 1. There, a classifier $\psi(h)$ predicts the class 1 in the orange regions of the input space $\mathcal{X}$, and class 0 otherwise. In the *large* separation regime that is captured by the family of distributions in $\mathcal{H}_1$, the orange regions are far apart and have a very small effect on each other (this is seen pictorially as each positive regions shrinks uniformly upon smoothing – as if there were no other positive regions). In these cases, training on noisy data leads to each of the orange regions to shrink in the final smoothed classifier $\mathrm{Smooth}_\beta(\psi(h * p_\alpha))$ for any $\alpha > 0$, and the smoothed classifier with noisy training has larger benign risk than training with no noise (Figure 1 top row). On the other hand, as the separation parameter decreases the orange regions get closer to each other eventually having a significant effect on their neighbors after smoothing (Figure 1 bottom row). In this regime, it becomes possible to set $\alpha > 0$ to achieve better benign risk after smoothing compared to smoothing a classifier not trained with noisy data (i.e. with $\alpha = 0$). Such distributions are captured in $\mathcal{H}_2$. For these distributions, the result in Theorem 3.2 is in fact not tight, and thus benefit can be derived from setting $\alpha > 0$.

With these informal results in mind, we now present them with greater rigour in the next section, before moving to the numerical experiments later in Section 5.

## 4    Detailed Results

In this section we will expand and make our results from Section 3 more precise. We will firstly prove in Section 4.1 the phenomenon observed in the *large separation* regime illustrated in the top panel of Figure 1, and demonstrated in our experiments (Figure 3). Next, in Section 4.2, we will construct data distributions that have a specific structure, following the phenomenon observed in the *small separation* regime, illustrated in the bottom panel of Figure 1.

Given a conditional distribution $h$, we define a partition of the input space $\mathcal{X}$ as $\mathcal{X} = \mathcal{I}^- \cup \mathcal{I}^+$, where $\mathcal{I}^+$ contains all the points where the classifier outputs 1, i.e., $\mathcal{I}^+ = \{x \in \mathcal{X} : \psi(h)(x) = 1\}$. Further, we think of $\mathcal{I}^+$ as being composed of disjoint *positive* partitions $I_1, I_2, \ldots, I_M$ such that each partition is simply connected [6]. $\mathcal{I}^-$ is the rest of the space, i.e., $\mathcal{X} \setminus \mathcal{I}^+$, and is partitioned analogously. In other words,

$$\mathcal{I}^+ = \mathcal{I}_1 \sqcup \mathcal{I}_2 \sqcup \ldots \sqcup I_M \text{ and } \mathcal{I}^- = \mathcal{X} \setminus \mathcal{I}^+,$$

where $\sqcup$ denotes the disjoint union. Note that how to exactly obtain a valid partition above is unspecified – we construct the partitions explicitly for Theorems 4.1 and 4.3, and Theorem 4.2 holds for any valid partitioning. In Figure 1, these regions are colored orange. When a classifier is trained on data perturbed by noise $p_\alpha$, and then smoothed by noise $p_\beta$, each of these positive partitions change in some way. Denote these latter partitions $I_1(\alpha, \beta), \ldots, I_M(\alpha, \beta)$. In the following two subsections, we will show how the difference between

---

[6]A subset of the space $\mathcal{X}$ is defined in the standard topological sense to be simply connected, if it is a connected region containing no *holes*, i.e., every curve can be continuously contracted to a point.

$I_k$ and $I_k(\alpha, \beta)$ changes under different separation regimes, and how this in turn determines the behavior of $\Delta_{\alpha,\beta}$.

## 4.1 Large Separation Regime

We will first formalize and prove a version of Theorem 3.1 by constructing suitable distributions $\mathcal{H}_1$ with a large interference distance. We will then generalize the proof strategy to arbitrary distributions to obtain Theorem 3.2.

Unless specified otherwise, we assume that the noise distributions (i.e., those for smoothing) $p_\theta(x)$ we are working with are *nice*, in the following specific sense. Nice probability distributions are decreasing in their argument $x$, and are spherically symmetric. Formally, a probability distribution $p$ over $\mathbb{R}^d$ parameterized by a scalar $\alpha \in \mathbb{R}$ is defined to be *nice*, if for all $x_1, x_2 \in \mathbb{R}^d$, $p$ satisfies the properties

$$\text{(Decreasing in argument norm)} \quad p_\alpha(x_1) \geq p_\alpha(x_2), \quad \text{if } \|x_1\|_2 \leq \|x_2\|_2, \text{ and,}$$
$$\text{(Spherically symmetric)} \quad p_\alpha(x_1) = p_\alpha(x_2), \quad \text{if } \|x_1\|_2 = \|x_2\|_2.$$

Nice distributions comprise Uniform and Gaussian as special cases. Throughout the paper, we consider the *family* of $p_\alpha$ and $p_\beta$ to be the same (e.g., both Uniform or both Gaussian). This is not required for our analyses, and simple extensions of our results could generalize further to these having different forms.

The *lower* interference distance for a given $h$ is now defined as the minimum distance between any two positive partitions, i.e., $\underline{\zeta}_h = \min_{i \neq j} \text{dist}(I_i, I_j)$. The lower interference distance over a family $\mathcal{H}$ is defined as $\underline{\zeta}_\mathcal{H} = \min_h \underline{\zeta}_h$. We can now state the detailed version of Theorem 3.1. The full proof, along with further properties of these nice distributions, can be found in Appendix D.

**Theorem 4.1.** *For nice noise distributions $p_\alpha = \text{Unif}(B_{\ell_2}(0, \alpha))$ and $p_\beta = \text{Unif}(B_{\ell_2}(0, \beta))$, there exists $\mathcal{H}_1$ with interference distance $\underline{\zeta}_{\mathcal{H}_1} > \max(\alpha, \beta)$, such that for all $h \in \mathcal{H}_1$ we have $\Delta_{0,\beta}(h) < \Delta_{\alpha,\beta}(h)$ for all $0 < \alpha < \underline{\zeta}_{\mathcal{H}_1}$ and all $0 < \beta < \underline{\zeta}_{\mathcal{H}_1}$.*

While we tightly bound $\Delta_{\alpha,\beta}$ from above and below in Theorem 4.1, we are able to do so by assuming that the positive partitions $\mathcal{I}^+$ are perfectly spherical. In the following result, we show that we can in fact generalize our proof technique to obtain an upper-bound for $\Delta_{\alpha,\beta}$ in Theorem 4.2 that only assumes that $h$ is supported on the (bounded) data domain $\mathcal{X}$, with partitions of arbitrary shape. In doing so, we obtain an upper-bound that might be loose in general. However, we note that there exist simple distributions ($\mathcal{H}_1$) where the upper-bound in Theorem 4.2 is tight, implying that it is the best one could hope for without assuming anything else about $h$. The full proof can be found in Appendix E.

**Theorem 4.2.** *For nice noise distributions $p_\alpha, p_\beta$, for any $h$ supported on a bounded domain $\mathcal{X}$,*

$$\Delta_{\alpha,\beta}(h) \leq 1 - \sum_k p_X(B_{\ell_2}(\hat{x}^k, (\omega_{h,\tau}^k - r_{\alpha,\beta}^k)_+)), \tag{3}$$

*where $r_\alpha^k = \sqrt{\Psi_\alpha^{-1}\left(\frac{0.5}{0.5+\tau}\right)}$ if $\mathcal{I}_k$ is a positive partition, i.e., $\mathcal{I}_k \in \mathcal{I}^+$ and $r_\alpha^k = \sqrt{\Psi_\alpha^{-1}\left(\frac{0.5}{0.5-\tau}\right)}$ otherwise, i.e., $\mathcal{I}_k \in \mathcal{I}^-$. Further, $r_{\alpha,\beta}^k = r_\alpha^k + \sqrt{\Psi_\beta^{-1}(0.5)}$, and $\Psi_\alpha, \Psi_\beta$ are the CDFs of the distribution of $\|z\|_2^2$ when $z \sim p_\alpha$ and $z \sim p_\beta$, respectively. $p_X(S)$ denotes the measure of the set $S$ under $p_X$. The upper bound (3) holds for any choice of $\{(\omega_{h,\tau}^k, \hat{x}^k)\}$ such that $\hat{x}^k$ is the center of a ball of radius $\omega_{h,\tau}^k$ completely contained in $\mathcal{I}_{k,\tau}$ defined as $\mathcal{I}_{k,\tau} = \{x \in \mathcal{I}_k: |h(x) - 0.5| \geq \tau\}$. We can choose the sequence $\{(\omega_{h,\tau}^k, \hat{x}^k)\}$ such that the upper bound (3) is minimized.*

This results upper bounds the excess risk by one minus the sum of the measures of balls under the data distribution. Each partition $\mathcal{I}_k$ contributes to the upper bound in (3) via the difference $(\omega^k - r^k)$. The first term $\omega_{h,\tau}^k$ denotes the *inradius* of a subset of $\mathcal{I}_k$, defined as the portion of $\mathcal{I}_k$ classified with a confidence margin $\tau$, i.e., $\mathcal{I}_{k,\tau} = \{x \in \mathcal{I}_k: |h(x) - 0.5| \geq \tau\}$. The second term $r^k$ captures the shrinkage produced by smoothing. Specifically, $r_\alpha^k$ denotes the shrinkage caused in $\mathcal{I}_k$ while moving from the original classifier to the

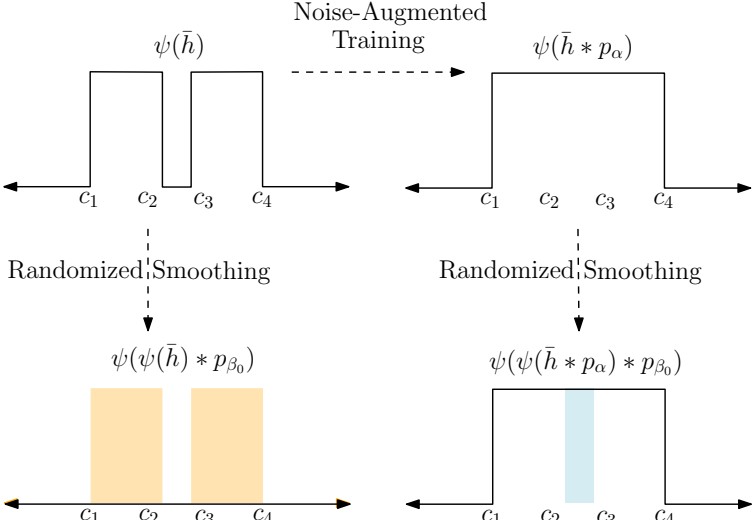

Figure 2: An example $\bar{h} \in \mathcal{H}_2$ where noise-augmented training is beneficial for randomized smoothing. The orange and blue shaded bars denote the regions where the smoothed classifiers make errors, i.e., $\psi(\bar{h}) \neq \text{Smooth}_{\beta_0}(\psi(\bar{h}))$ and $\psi(\bar{h}) \neq \text{Smooth}_{\beta_0}(\psi(\bar{h} * p_{\alpha_0}))$ respectively. Assuming a suitable data-distribution $p_X$, the orange region can have a higher $p_X$-mass than the blue region, i.e., $\Delta_{\alpha_0, \beta_0}(\bar{h}) < \Delta_{0, \beta_0}(\bar{h})$. See the text below for details of the four sub-figures.

noise-trained classifier, i.e., $\psi(h) \to \psi(h * p_\alpha)$. Similarly, $r^k_{\alpha, \beta}$ denotes the shrinkage caused while moving from the noise-trained classifier to the randomized smoothed classifier, i.e., $\psi(h * p_\alpha) \to \psi(\psi(h * p_\alpha) * p_\beta)$. For any fixed $\beta$, as we increase the noise-augmentation strength $\alpha$, the shrinkage $r^k_{\alpha, \beta}$ also increases, leading to an increase in the upper bound in Eq. (3).

## 4.2 Small Separation Regime

We will show that, unlike what is reflected by Theorem 4.1, there exists a family of distributions $\mathcal{H}_2$ characterized by a small separation distance where $\Delta_{\alpha, \beta_0}$ is indeed minimized at some $\alpha > 0$, which is the behavior observed in practice in common image datasets.

We will begin with an illustrative one-dimensional example of a data distribution supported on $\mathcal{X} \subset \mathbb{R}$ where we will show that the interference distance $\zeta$ explicitly controls whether any benefit can be obtained by noise-augmentation. We will observe the basic structure required in $h$ for this to occur, and then generalize this structure to construct examples supported on $\mathcal{X} \subset \mathbb{R}^d$ forming the family $\mathcal{H}_2$ required for Theorem 4.3.

$1-$**dimensional example** We will now walk through the four panels of Figure 2. First, we let $\bar{h}$ be defined as

$$\bar{h}(x) = \begin{cases} 0 & x \in (-0.5, c_1) \cup (c_2, c_3) \cup (c_4, 0.5) \\ 1 & \text{otherwise,} \end{cases} \tag{4}$$

where the specific values of $c_1, \dots c_4$ are not important for the example but can be found in Appendix F. Additionally, let $p_X$ be uniform in $[c_1, c_4]$ – this assumption is not needed, but simplifies the following discussion. The resulting $\psi(\bar{h})$ is plotted in the top-left panel of Figure 2.

For smoothing, we use the nice distribution $p_{\beta_0} = \text{Unif}([-\beta_0/2, \beta_0/2])$ where $\beta_0$ is just large enough, so that $(\psi(\bar{h}) * p_{\beta_0})(x) = 0$ for all $x \in [c_1, c_4]$. In other words, randomized smoothing ensures that the smoothed classifier $\text{Smooth}_{\beta_0}(\psi(\bar{h}))$ outputs a consistent label across the input, hence ensuring robust classification (and in particular, that label is 0). However, observe that the benign risk of the smoothed classifier $R(\text{Smooth}_{\beta_0}(\psi(h)))$ is very high, as it makes an error whenever $x \in [c_1, c_2] \cup [c_3, c_4]$. This situation is shown in the bottom left panel of Figure 2.

We now consider the nice noise distribution for noise-augmented training, defined as $p_{\alpha_0} = \text{Unif}([-\alpha_0/2, \alpha_0/2])$. For a suitable value of $\alpha$, we can ensure that $\psi(\bar{h} * p_{\alpha_0})(x) = 1$ for all $x \in [c_1, c_4]$. This is shown in the top right panel of Figure 2. But now, since $\beta_0$ was just enough to obtain $\text{Smooth}_{\beta_0}(\psi(\bar{h})) = 0$, the additional positive mass in $[c_2, c_3]$ causes the classification to flip, and now $\text{Smooth}_{\beta_0}(\psi(\bar{h}) * p_{\alpha_0})(x) = 1$ for all $x \in [c_1, c_4]$. The benign risk of the smoothed classifier now is much lower than earlier, as it only makes an error on the small crevice $[c_2, c_3]$. This situation is shown in the bottom right panel of Figure 2. Thus, one can choose $\bar{h}, p, \alpha_0, \beta_0$ such that

$$R(\text{Smooth}_{\beta_0}(\psi(\bar{h}) * p_{\alpha_0})) < R(\text{Smooth}_{\beta_0}(\psi(\bar{h}))), \tag{5}$$

demonstrating that training with noise augmentation is indeed beneficial when $\bar{h}$ has a low interference distance. The specific values of $\alpha_0, \beta_0$ can be found in Appendix F.

Fixing the noise-augmentation distribution $p_{\alpha_0}$ and the smoothing distribution $p_{\beta_0}$, we now modify $\bar{h}$ to $\tilde{h}$ by increasing the interference distance $|c_2 - c_3|$ while maintaining the same structure, i.e. $\tilde{h}(x) = 0$ when $x \in (-0.5, \tilde{c}_1) \cup (\tilde{c}_2, \tilde{c}_3) \cup (\tilde{c}_4, 0.5)$, and $\tilde{h}(x) = 1$ otherwise. When $|\tilde{c}_2 - \tilde{c}_3| > \alpha_0/2$, noise-augmentation with $p_{\alpha_0}$ is no longer effective, i.e., $\psi(\psi(\tilde{h}) * p_{\alpha_0}) = \psi(\psi(\bar{h}) * p_{\alpha_0})$. Intuitively, this is caused by insufficient positive mass near any class-0 point for the prediction to flip after noise-augmentation – and this can be verified using the values of $\alpha_0, \beta_0$ provided in Appendix F). Applying randomized smoothing on $\psi(\psi(\tilde{h}) * p_{\alpha_0})$ now simply leads to $\text{Smooth}_{\beta_0}(\psi(\tilde{h}) * p_{\alpha_0})(x) = 0$ everywhere, implying that the risk after smoothing is high, i.e.,

$$R(\text{Smooth}_{\beta_0}(\psi(\tilde{h}) * p_{\alpha_0})) \geq R(\text{Smooth}_{\beta_0}(\psi(\tilde{h}))). \tag{6}$$

Thus, we have shown via this example that the interference distance directly controls whether we get any benefit out of training with noise-augmentation (5) or not (6).

The distance $|c_2 - c_3|$ above corresponds to a small *upper* interference distance $\bar{\zeta}$, which we define now as the maximum distance between any two positive partitions, i.e., $\bar{\zeta} = \max_{i \neq j} \text{dist}(I_i, I_j)$. With this definition, we can now extend the ideas in the simple example above to general constructions in $d-$dimensions.

**Theorem 4.3.** *There exist nice distributions $p_\alpha, p_\beta$, a family $\mathcal{H}_2$ with low interference distance $\overline{\zeta_{\mathcal{H}_2}}$, and data-distributions $p_X$, such that for all $h \in \mathcal{H}_2$ we have $\Delta_{0,\beta_0}(h) > \Delta_{\alpha,\beta_0}(h)$ for some $\alpha > 0, \beta_0 > 0$.*

We pause again to understand some implications of Theorem 4.3. From the one-dimensional example and the proof of Theorem 4.3, we see that training with noisy data, i.e., $\alpha > 0$, is better for distributions with low interference distance. We conjecture that natural distributions follow this structure as well, containing regions corresponding to one class that are packed closely in the domain. Interestingly, we find evidence in the favor of this conjecture in our experiments in Section 5. Moreover, this result also shows that there is no reason for the noise level $\alpha$ to be the same as the smoothing level $\beta$ for final smoothed classifier to obtain its lowest risk. This suggests that the common practice of using $\alpha = \beta$ for randomized smoothing (e.g., see (Cohen et al., 2019)) might not be optimal. Indeed, we will shortly demonstrate in our experimental section that better choices for smoothing can be found by relaxing the constraint that the training noise level $\alpha$ should be equal to $\beta$. Additionally, our findings might provide a theoretical foundation for recent works (Alfarra et al., 2020; Anderson and Sojoudi, 2022; Súkeník et al., 2021) on obtaining an *adaptive* $\beta$ for smoothing at each input point.

**Exactly Learning Bayes Classifiers**   We pause here to comment on the assumptions we made in analyzing randomized smoothing on a noise-smoothed distribution, $\text{Smooth}_\beta(\psi(h * p_\alpha))$. As we mentioned above – and as we prove in Appendix B – our analysis assumed that one has access to the true conditional distribution of $\mathbb{E}[Y|X_s]$. However, in more realistic settings, the learned predictor might deviate from this optimal value. We now argue that our analysis is still relevant in these cases, too.

On the one hand, in Theorem G.1, we provide an extension of our result in Theorem 4.1 by considering a predictor that is an inexact approximation to the conditional expectation. In other words, we assume that training with noise-augmentation results in a function $g(x)$ that is not too far from the true smoothed conditional distribution, $|g(x) - h * p_\alpha(x)| \leq \eta$ for all $x \in \mathcal{X}$. As we show in Appendix G, our same proof technique allows us to show that indeed, there exist distributions where no benefit can be expected from

randomized smoothing, even in this more general case where the assumption of learning the true conditional distribution is relaxed.

On the other hand, it is also not hard to see that our results that provide an upper bound to the excess risk based on Bayes classifier, $\Delta_{\alpha,\beta}(h)$, are also useful in characterizing upper bounds to more realistic predictors that might deviate from the true conditional distribution. Indeed, given access to $g$ that is only an approximation to $h * p_\alpha$, one can show (see our derivation in Appendix G) that the excess risk of smoothing $g$ can be upper bounded by

$$\Delta_{\alpha,\beta}(g,h) \leq \Delta_{\alpha,\beta}(h) + p_X\left((h * p_\alpha)(X) \neq g(X)\right). \tag{7}$$

As a result, our upper bound in Theorem 4.2 also provides an upper bound to the excess risk of more general predictors $g$ – and the tightness of this bound is naturally controlled by the mass under $p_X$ of the errors between $g$ and $h * p_\alpha$.

**Extension to Multiple Classes**  Although our theory was developed for binary classification, a standard extension to $K$-way classification is possible, using the technique followed while certifying multi-class Randomized Smoothing based classifiers Cohen et al. (2019), Yang et al. (2020). In this context, the class $Y = 0$ now represents one of the $K$-classes, and $Y = 1$ represents all the remaining classes. All our theorems would then extend, having an additional parameter for the class viewed as 0.

Having established our theoretical results, we now turn to an empirical verification of some of them on synthetic datasets, as well as MNIST and CIFAR-10.

## 5 Experiments

**Synthetic Experiments.** We begin by conducting synthetic experiments[7] with distributions resembling those used in the proofs of our existence results Theorems 3.1 and 3.3. We take $\mathcal{X} = [0, 100] \times [0, 100]$ as the data domain, and sample the positive partitions of $h$ as spheres at random locations in the domain (see Appendix A for more details about the construction) to obtain the data-distribution $h^\zeta$, where the superscript $\zeta$ denotes the interference distance. Figure 3 shows a few examples of $h^\zeta$ for different values of $\zeta$.

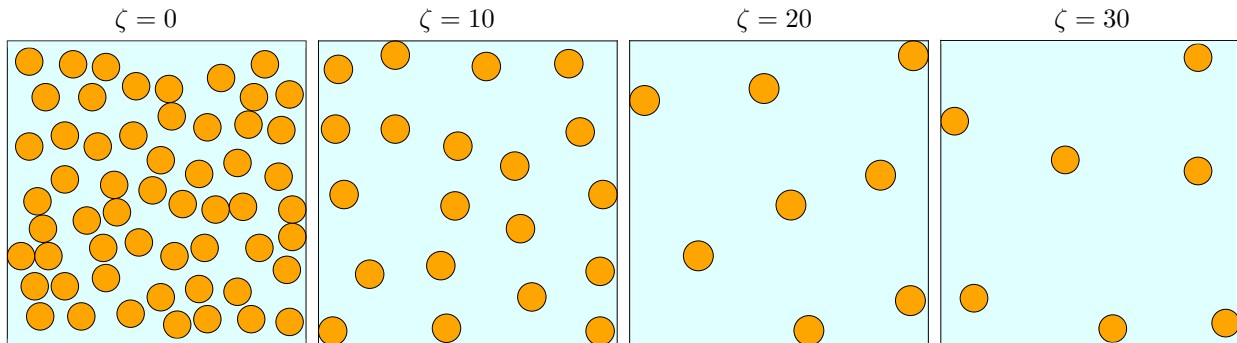

Figure 3: Examples of $\psi(h^\zeta)$ for different values of $\zeta$. The blue regions denote $\psi(h^\zeta)(x) = 0$ and the orange regions denote $\psi(h^\zeta)(x) = 1$.

In Figure 4, we report quantitative results using our synthetic distributions as described above. Over several choices of interference distance $\zeta$, we first sample several synthetic data distributions $h^\zeta$. We then plot the quantity $\Delta_{\alpha,\beta} - \Delta_{0,\beta}$, which is the difference between the benign risks of smoothing a noise-augmented classifier, i.e., $R(\text{Smooth}_\beta(\psi(h * p_\alpha)))$, and smoothing a classifier not trained with noise-augmentation, i.e., $R(\text{Smooth}_\beta(\psi(h)))$. We increase the smoothing strength $\beta$ moving from left to right in Figure 4, and plot the difference in the risks as a function of $\alpha$ for each $\beta$. Recall that noise-augmentation is useful only when we can

---

[7]We provide code for all of our experiments at https://github.com/ambarpal/randomized-smoothing.

find $\alpha_0, \beta_0$ such that $\Delta_{\alpha_0, \beta_0} - \Delta_{0, \beta_0} < 0$. Accordingly, we plot a dashed line whenever noise-augmentation is useful, i.e., $\exists \alpha_0, \beta_0 \ \Delta_{\alpha_0, \beta_0} < \Delta_{0, \beta_0}$ and solid otherwise, i.e., $\forall \alpha, \beta \ \Delta_{\alpha, \beta} \geq \Delta_{0, \beta}$.

At high $\zeta$, we are operating in the *large interference distance* regime and the results from Section 4.1 apply. Those results dictate that $\Delta_{\alpha, \beta_0}$ should be minimized at $\alpha = 0$ for any fixed $\beta_0 > 0$. Indeed, this phenomenon can be seen in Figure 4 for $\zeta = 3$, where we see that the line remains solid for all $\beta$.

On the other hand, at lower $\zeta$, we go into the *small interference distance* regime and the results from Section 4.2 apply. Those results dictate that $\Delta_{\alpha, \beta_0}$ should be minimized at some $\alpha > 0$ given a large enough, but fixed $\beta_0 > 0$. Indeed this phenomenon is seen in Figure 4, where the lines tend to become dashed as $\beta$ increases. Thus, our synthetic experiments confirm the predictions from our theory.

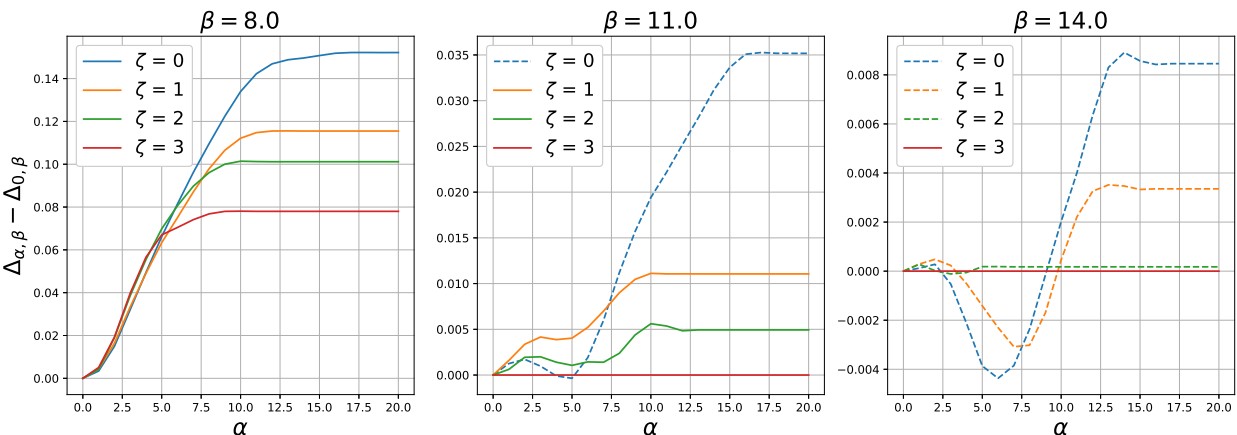

Figure 4: Synthetic Experiments showing the variation in $\Delta_{\alpha, \beta}$. The line is dashed if there exists $\alpha > 0$ such that $\Delta_{\alpha, \beta} < \Delta_{0, \beta}$, and solid otherwise.

**MNIST, CIFAR-10 Experiments.** We now proceed to conduct experiments with real data distributions. For a range of finely spaced $\alpha$ in $[0, 1]$, we train a standard CNN $h_\alpha$ with isotropic Gaussian noise-augmentation with variance $\alpha^2$, i.e., $p_\alpha = \mathcal{N}(0, \alpha^2 I)$. For each of these trained models, we use the isotropic-Gaussian with variance $\beta^2$, with $\beta \in [0, 1]$, as the smoothing distribution, i.e., $p_\beta = \mathcal{N}(0, \beta^2 I)$, to obtain the smoothed classifier $\text{Smooth}_\beta(\psi(h * p_\alpha))$. We then plot the standard empirical estimate of the risk, $\hat{\Delta}_{\alpha, \beta} = \sum_{(x, y) \in S_{\text{test}}} \mathbb{1}[\text{Smooth}_\beta(h_\alpha)(x) \neq y]$ over the test set $S_{\text{test}}$, for different $\alpha, \beta$ in Figure 5.

The first observation that we make from the MNIST and CIFAR-10 plots in Figure 5 is that a non-zero data-augmentation is always beneficial, i.e., $\alpha^* > 0$. While not an implication arising from our theory, this behavior is reminiscent of our results in Theorem 4.3, suggesting that real data distributions lie in the *small interference distance* regime. Additionally, we note the similarity of the synthetic experiment plots at low $\zeta$ in Figure 4 and the real-data curves in Figure 5 (upto scaling of the $\alpha$-axis, and the fact that we subtract $\Delta_{0, \beta}$ for normalization in Figure 4), providing empirical validation for our theoretical assumptions.

The second observation we make from Figure 5 is that, as suggested by Theorem 4.3, the smoothing parameter need not be the same as the data augmentation parameter for the best performance of the smoothed classifier. This phenomenon has also been observed empirically in prior work Alfarra et al. (2020), Salman et al. (2019), Zhai et al. (2020), and our results provide a theoretical foundation for such observations.

## 6 Related Work

In this work, we study randomized smoothing, which is currently the method of choice for obtaining robustness guarantees for deep, complex neural network classifiers. Since the early proposals of stochastic smoothing (Liu et al., 2018), many works have derived robustness *certificates* against adversarial attacks for smoothed classifiers (Lecuyer et al., 2019; Li et al., 2018; Cohen et al., 2019). These certificates obtain a radius $r$ around an input $x$ such that the network prediction is robust in an $\ell_p$ ball of radius $r$ around $x$. Studied

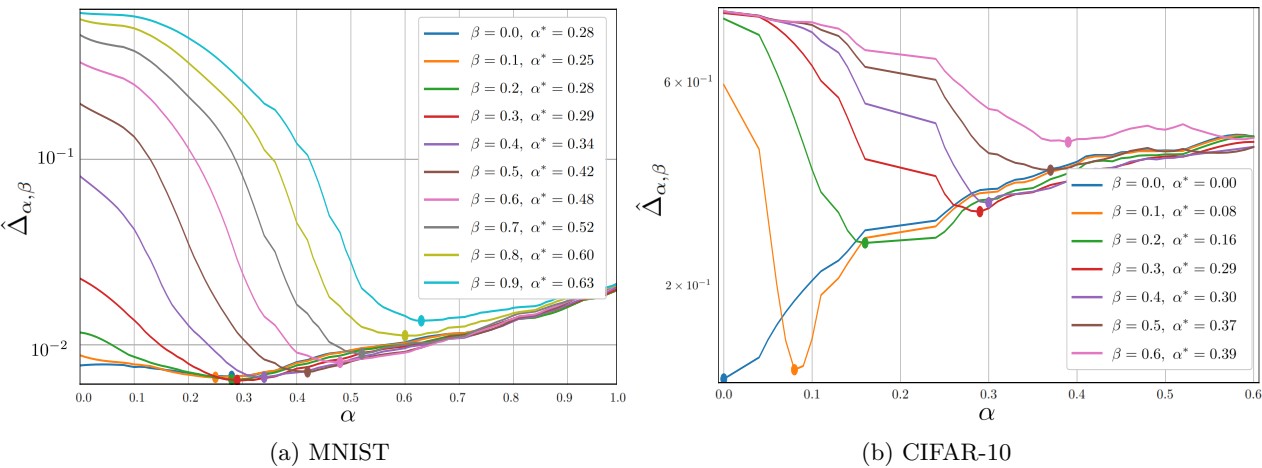

Figure 5: Plot of $\hat{\Delta}_{\alpha,\beta}$ for MNIST (5a) and CIFAR-10 (5b). For each fixed value of $\beta = \beta_0$, the legend $\alpha^*$ (and the corresponding shaded disk on the plot) shows the minimizer of the $\Delta_{\alpha,\beta_0}$ curve.

balls include $p = 0$ (Levine and Feizi, 2020a; Lee et al., 2019), $p = 1$ (Teng et al., 2019), $p = 2$ (Cohen et al., 2019; Salman et al., 2020; 2019) and $p = \infty$ (Zhang, 2002). Certificates have also been extended to non-$\ell_p$ smoothing distributions (Levine and Feizi, 2020b). For different notions of optimality, the optimal certificates have also been characterized (Yang et al., 2020; Kumar et al., 2020).

However, there has been lesser attention given to the impact that randomized smoothing has on the benign accuracy of the smoothed classifier, and a theoretical understanding for the need—or lack thereof—for noise-augmented training of the classifier in order to produce an accurate smoothed classifier. Recent work in Gao et al. (2020), which is closest to ours in spirit, takes a first step towards this goal by analyzing the class of hypotheses that are realizable when training on smoothed data. Their result shows that this re-training strictly reduces the hypothesis class whenever the smoothing strength is above a threshold. In this way, Gao et al. (2020) provide some theoretical basis for the reduction of accuracy of the so smoothed-trained classifier. Elsewhere, Mohapatra et al. (2021) theoretically demonstrate that randomized smoothing leads to shrinkage of class boundaries for simple data distributions. Importantly, their definition of "shrinkage" is that the bounding sphere (or cone, for "semi-bounded" regions) of the decision boundary becomes smaller. This is different from our analysis, as a shrunk decision region $R_\sigma$ might not be a subset of the original region $R$ under this definition. In comparison, our universal result in Theorem 4.2 holds in much more generality for arbitrary (bounded) data-distributions, and our existence results Theorems 4.1 and 4.3 construct specific synthetic datasets. While Mohapatra et al. (2021) only analyse the distance $\|f - \text{Smooth}_\beta(f)\|_2$ given a trained classifier $f$, we directly bound the risk $R(\text{Smooth}_\beta(f))$, and shrinkage does show up in some parts of our analysis. Our proof techniques are able to handle both noise-augmentation of strength $\alpha$ and randomized smoothing of strength $\beta$ simultaneously, allowing us to discover cases where noise-augmentation helps randomized smoothing (in other words, showing cases where the combined effect might not be a shrinkage of the decision boundaries). Some recent works have tried to move away from having the same smoothing and noisy-training distribution, by studying input-dependent smoothing (Alfarra et al., 2020; Anderson and Sojoudi, 2022; Súkeník et al., 2021). Lastly, and in a very different context of graph convolutional networks (GCNs), recent results in Keriven (2022) show that a positive level of smoothing (defined as the number of GCN layers) can be beneficial for a supervised learning task, before becoming detrimental once smoothing is too large. Studying further connections between our analysis and smoothing for GCNNs constitutes an interesting direction of research.

## 7 Conclusion, Limitations and Future Work

In this work, we provided a theoretical understanding of how noise-augmented training affects classifiers defended using randomized smoothing. We identified a parameter of the data distribution, which we dubbed *interference distance*, that, for certain families of data-distributions, determines whether noise-augmented training could be beneficial for randomized smoothing. Using this parameter, we showed there exist data distributions where noise-augmented training reduces the accuracy of the final smoothed classifier, contrary to common observation. Our upper bound to the benign risk in Theorem 4.2 for very general conditional distributions – and which is tight for distributions presented in Theorem 4.1– demonstrates that no benefit of randomized smoothing is possible without further structural distributional assumptions. In turn, we showed the existence of distributions with small interference distance in Theorem 4.3, where improvements by noise-augmented training is possible. We complemented our theoretical results with empirical validation, suggesting that real-world distributions lie in regimes where the parameter $\zeta$ is small, and so noise-augmented training can indeed help randomized smoothing if the smoothing strength is chosen properly. Contrary to intuition, the proof of our theoretical result in Theorem 4.3, and our experiments, suggest that this smoothing strength need not be the same as the noise-augmentation strength for best performance of the final predictor.

We now revisit the open questions that we posed in Section 1. Firstly, why is noise-augmentation helpful while training the base classifier? We saw in the bottom row of Figure 1 that noise augmented training helps the smoothed classifier have good benign accuracy when we are in the low-interference distance regime, as it alleviates the degradation of the risk incurred by randomized smoothing at deployment time. Secondly, how does the performance of the smoothed classifier depend on the training noise distribution? We saw in Theorems 4.1 and 4.3 that in the high interference regime, training with low to zero noise leads to good performance of the smoothed classifier. On the other hand, in the low-interference regime, the training noise can be tuned to extract good performance of the smoothed classifier. These conclusions are derived for the cases when the same family of distribution is used for augmentation during training and for smoothing at deployment time. Future work could consider extending these to different classes of distribution. Thirdly, is noise-augmented training needed for all data-distributions? The answer is no, noise-augmentation is not always effective in reducing the risk of the classifier after randomized smoothing, and the interference-distance parameter can distinguish between families of data-distributions where noise-augmentation helps randomized smoothing. It remains unclear how to efficiently determine this parameter for real data-distributions, which would allow us to determine whether noise-augmented training is needed at all, or what the optimal parameters of the augmentation distribution should be. This remains matter of future research.

We note that our results are independent of the learning aspect of the problem, i.e., we assume that the learning algorithm is good enough to obtain the predictor with the lowest possible risk. These assumptions, while simplistic, nonetheless allowed us to characterize observations that have practical relevance. Though we provide some results on how to relax such assumptions in Appendix G, we see this as a starting point for future work that should explore these trade-offs in more generality. Additionally, our experiments and theory suggest that the best data-augmentation strength is not the same as the smoothing strength in general, and we hope that future work can theoretically quantify and provide precise estimates for this quantity. Finally, in this paper, we studied the benign risk of classifiers after smoothing. Future work could extend these results to the study of the robust risk (which is lower bounded by the benign risk), and extend our techniques to quantify how the former depends on the noise-augmentation distribution.

**Acknowledgements**

The authors thank René Vidal for insightful comments, as well as the anonymous reviewers for their valuable suggestions, which helped improve this manuscript. AP acknowledges the support of DARPA (via GARD HR00112020010) and NSF (via Grant 1934979). JS acknowledges support from DARPA GARD HR00112020004.

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

## A    Experimental Details

**Synthetic Experiments**    To construct our synthetic distributions, we take $\mathcal{X} = [0, 100] \times [0, 100]$ as the data domain. We start with an empty set $\mathcal{I} = \{\}$ and radius $r = 10$. We repeat the following steps 500 times: (1) Sample a center $c$ in $\mathcal{X}$ uniformly at random. (2) Test whether $\|c - c'\|_2 > \zeta + 2r$ for all spheres $B_{\ell_2}(c', r) \in \mathcal{I}$. If true, add $B_{\ell_2}(c', r)$ to $\mathcal{I}$, otherwise reject this sample. At the end of this process, we obtain a set of spheres $\mathcal{I}$ satisfying the property $\|c - c'\|_2 \geq \zeta + 2r$ for all $c \neq c'$, where $\zeta$ is the interference distance.

**MNIST Experiments**    We take 100 samples from each of $p_\alpha, p_\beta$ per image $x$ in the MNIST test set to compute a single point $\Delta_{\alpha,\beta}$. The figures are quite stable against random initialization due to the MNIST test set size of 10000 samples. We use the following architecture for the neural network classifier $h_\alpha$:

$$\text{Input}(28, 28) \rightarrow \text{Conv}_{3,1}(1, 32) \rightarrow \text{Conv}_{3,1}(32, 64) \rightarrow \text{Linear}(9216, 128) \rightarrow \text{Linear}(128, 2)$$

In the above, the input image has dimension $28 \times 28$, and convolution layers with a filter size of 3 and a stride of 1 are applied. The notation $\text{Conv}_{3,1}(a, b)$ denotes that the input depth is $a$ and the number of filters is $b$ for the convolution. Finally, the notation $\text{Linear}(a, b)$ denotes a affine transformation with input dimension $a$ and output dimension $b$.

**CIFAR-10 Experiments**    We use the DLA Architecture (Yu et al., 2018) for the neural network classifier. For obtaining $h_\alpha$, we use standard noise augmentation on the CIFAR-10 dataset, i.e., we augment every training sample with noise sampled from $p_\alpha$. For obtaining the smoothed classifier $\text{Smooth}_\beta(h_\alpha)$, we use 200 noise samples from $p_\beta$ for every image. We report results aggregated over 4000 images from the CIFAR-10 test set.

## B    Derivation of Bayes Classifiers

Recall that $X$ and $Y$ denote the data and label random variables, respectively, such that $(X, Y) \sim p_{X,Y}$. In this binary setting, the Bayes classifier, i.e. the classifier that minimizes the probability of misclassification, can be obtained by thresholding the conditional expectation $\mathbb{E}[Y|X = x]$ at 0.5. We assume that the learning algorithm results in a predicted score $h(x) = \mathbb{E}[Y|X = x]$. The prediction of this model is given by $\psi(h)$.

Now, we instead train on noisy data, $X_s \sim p_X^s$, where $X_s = X + V$ such that $V \sim p_V$. It is well known that $p_X^s = p_X * p_V$, where $*$ denotes the convolution operator (Larsen and Marx, 2005). Further, we can show now that the conditional expectation is $\mathbb{E}[Y|X_s = x] = h * p_V$:

$$
\begin{aligned}
\mathbb{E}[Y|X_s = x] &= p(Y = 1|X_s = x) = p(Y = 1|X + V = x) \\
&= \int_v p(Y = 1|X + V = x, V = v)p_V(v)dv \\
&= \int_v p(Y = 1|X = x - v, V = v)p_V(v)dv \\
&= \int_v p(Y = 1|X = x - v)p_V(v)dv \qquad (8) \\
&= \int_v h(x - v)p_V(v)dv = (h * p_V)(x),
\end{aligned}
$$

where (8) uses the fact that the noise variable $V$ is independent of the data variable $X$. As earlier, the Bayes classifier is now given by a thresholding of the conditional expectation, i.e., $\psi(h * p_V)$.

## C Nice Distributions and their Properties

*Nice Distributions.* A probability distribution $p$ over $\mathbb{R}^d$ parameterized by a scalar $\alpha \in \mathbb{R}$ is defined to be *nice*, if $p$ satisfies the properties (9) and (10):

$$\text{(Decreasing in argument norm)} \qquad p_\alpha(x_1) \geq p_\alpha(x_2), \quad \text{if } \|x_1\|_2 \leq \|x_2\|_2 \tag{9}$$

$$\text{(Spherically symmetric)} \qquad p_\alpha(x_1) = p_\alpha(x_2), \quad \text{if } \|x_1\|_2 = \|x_2\|_2 \tag{10}$$

For ease of understanding of the rest of the section, one can think of $p$ to be the uniform distribution supported on a ball of radius $\theta$, i.e., $p_\theta = \text{Unif}(B_{\ell_2}(0, \theta))$, which is a nice distribution.

*Shifted CDF.* We define the multivariate cumulative distribution function (CDF) w.r.t. $x \in \mathcal{X} \subset \mathbb{R}^d$ as

$$\Phi_\alpha(r, x) = \int_{\|t\|_2 \leq r} p_\alpha(x - t) dt \tag{11}$$

For nice distributions $p$, we can show the following properties for the CDF:

$$\text{(Decreasing in argument Norm)} \qquad \Phi_\alpha(r, x_1) \leq \Phi_\alpha(r, x_2), \quad \text{if } \|x_1\|_2 \geq \|x_2\|_2 \tag{12}$$

$$\text{(Spherically symmetric)} \qquad \Phi_\alpha(r, x_1) = \Phi_\alpha(r, x_2), \quad \text{if } \|x_1\|_2 = \|x_2\|_2 \tag{13}$$

$$\text{(Increasing in radius)} \qquad \Phi_\alpha(r_1, x) \geq \Phi_\alpha(r_2, x), \quad \text{if } r_1 \geq r_2 \tag{14}$$

To see (13), we choose any $x_1$, $x_2$ with $\|x_1\| = \|x_2\|$. Now, note that the integral in (11) can be rewritten as $\Phi_\alpha(r, x) = \int_{t \in B_{\ell_2}(x,r)} p_\alpha(t) dt$, which in other words computes the mass of the $p_\alpha$ contained in an $\ell_2$ ball of radius $r$ around $x$. Since $\|x_1\| = \|x_2\|$, we can find a rotation matrix $Q$ that maps $x_1$ to $x_2$ as $Qx_1 = x_2$. Observe that the same rotation matrix $Q$ also maps the entire ball $B(x_1, r)$ to the ball $B(x_2, r)$. Finally, (13) follows from by observing that $p_\alpha(x) = p_\alpha(Qx)$ for any rotation matrix $Q$ as $\|x\|_2 = \|Qx\|_2$.

Now that we know that $\Phi_\alpha(r, x)$ is spherically symmetric in the second argument, we can focus our attention to the restriction of $\Phi$ along the standard basis vector $e_1$ for showing (12). For any $r > 0$, consider the function $\phi \colon \mathbb{R} \to \mathbb{R}$ defined as $\phi(c) = \Phi(r, ce_1)$ for a scalar $c$. Points $x_1$, $x_2$ with $\|x_1\| \geq \|x_2\|$ correspond to $c_1 = \|x_1\|$ and $c_2 = \|x_2\|$ in the sense that $\Phi(r, x_1) = \phi(c_1)$ and $\Phi(r, x_2) = \phi(c_2)$, and $c_1 \geq c_2 \geq 0$. Define $D_1 = B_{\ell_2}(c_1 e_1, r)$ and $D_2 = B_{\ell_2}(c_2 e_1, r)$. We have,

$$
\begin{aligned}
\phi(c_1) &= \int_{t \in D_1} p_\alpha(t) dt \\
&= \int_{t \in (D_1 \cap D_2)} p_\alpha(t) dt + \int_{t \in (D_1 \setminus D_2)} p_\alpha(t) dt \\
&\leq \int_{t \in (D_1 \cap D_2)} p_\alpha(t) dt + \int_{t \in (D_2 \setminus D_1)} p_\alpha(t) dt \\
&= \phi(c_2).
\end{aligned}
\tag{15}
$$

To obtain the inequality in (15), observe that $D_1, D_2$ are spheres of the same radius with their centers $c_1 e_1, c_2 e_2$ such that $0 \leq c_2 \leq c_1$. This ensures that any point in $D_2 \setminus D_1$ has a lower $\ell_2$ norm than any point in $D_1 \setminus D_2$, which in turn means that the every term in the integral $\int_{t \in (D_2 \setminus D_1)} p_\alpha(t) dt$ is lower bounded by every term in the integral $\int_{t \in (D_1 \setminus D_2)} p_\alpha(t) dt$, using Property (9) of $p_\alpha$.

Due to properties (12) and (13), we see that $\Phi_\alpha(r, x)$ is non-increasing as $\|x\|_2$ increases. As a result, we can define the function $A_{\alpha,r} \colon \mathbb{R} \to \mathbb{R}$ as

$$A_{\alpha,r}(c) = \max_{\{x \,:\, \Phi_\alpha(r,x) \geq c\}} \|x\|_2, \tag{16}$$

which computes the maximum $\ell_2$ norm that $\|x\|_2$ can take before the CDF falls below $c$. This function will be useful to us later when we reason about how much does smoothing a classifier *contract* or *expand* its positive regions.

We firstly note that $A_{\alpha,r}$ is decreasing in $c$. This can be seen as increasing $c$ makes the constraint tighter in (16), and hence the optimal objective value can only decrease when we increase $c$.

Secondly, we note that $A_{\alpha,r}(\Phi_\alpha(r, x_0)) = \|x_0\|_2$, for any $x_0 \in \mathcal{X}$, $\alpha \geq 0$ and $r \geq 0$. This can be seen by observing that for $x = x_0$, we have $c_0 = \Phi_\alpha(r, x_0)$, and thus $x_0$ is a feasible point for the optimization problem $\max_{\{x : \Phi_\alpha(r,x) \geq c_0\}} \|x\|_2$. Hence $A_{\alpha,r}(c_0) \geq \|x_0\|_2$. Then, by property (12), we know that $\Phi_\alpha(r, x') \leq \Phi_\alpha(r, x_0)$ whenever $\|x'\|_2 \geq \|x_0\|_2$, implying that any such $x'$ will not be feasible. This shows that $A_{\alpha,r}(c_0) \leq \|x_0\|_2$, completing the argument.

# D  Proof of Theorem 4.1

**Theorem 4.1.** *For nice noise distributions $p_\alpha = \mathrm{Unif}(B_{\ell_2}(0, \alpha))$ and $p_\beta = \mathrm{Unif}(B_{\ell_2}(0, \beta))$, there exists $\mathcal{H}_1$ with interference distance $\underline{\zeta}_{\mathcal{H}_1} > \max(\alpha, \beta)$, such that for all $h \in \mathcal{H}_1$ we have $\Delta_{0,\beta}(h) < \Delta_{\alpha,\beta}(h)$ for all $0 < \alpha < \underline{\zeta}_{\mathcal{H}_1}$ and all $0 < \beta < \underline{\zeta}_{\mathcal{H}_1}$.*

*Proof.* Recall that $h(x) = p(Y = 1|X = x)$. Let $I_1, I_2, \ldots, I_k$ be spheres such that $I_j = B_{\ell_2}(c_j, r_j)$. Define $h(x) = 0.5 + \tau$ whenever $x \in I_1 \cup I_2 \ldots \cup I_k$, and 0 otherwise.

**Large Interference Distance**   Recall that $\underline{\zeta}_h = \min_{i \neq j} \mathrm{dist}(I_i, I_j)$, and assume that $\underline{\zeta}_h > \max(\alpha, \beta)$.

Recall the CDF $\Phi_\alpha(x, r) = \int_{\|t\|_2 \leq r} p_\alpha(x - t)dt$, which is a function decreasing monotonically in $\|x\|_2$. Also recall the function $A_{\alpha,r}$ with the property $A_{\alpha,r}(\Phi_\alpha(x, r)) = \|x\|_2$ for all $x$. For any $x \in I_j$, we can obtain $h * p_\alpha$ as

$$
\begin{aligned}
h * p_\alpha(x) &= \int_{I_j} h(t)p_\alpha(x - t)dt + \int_{\mathcal{X} \setminus I_j} h(t)p_\alpha(x - t)dt \\
&= \int_{I_j} (0.5 + \tau)p_\alpha(x - t)dt + \int_{\mathcal{X} \setminus I_j} h(t) \cdot 0 \; dt \\
&= (0.5 + \tau)\Phi_\alpha(x - c_j, r_j).
\end{aligned}
$$

$\alpha-$**Shrinkage**   The regions where the classifier $\psi(h * p_\alpha)$ predicts 1 are given by $I_{j,\alpha} = \{x \in I_j : h * p_\alpha \geq 0.5\}$. This region $I_{j,\alpha}$ is an $\alpha$-shrinkage of $I_j$ and can be obtained as

$$
(0.5 + \tau)\Phi_\alpha(x - c_j, r_j) \geq 0.5 \implies \|x - c_j\|_2 \leq A_{\alpha,r_j}\left(\frac{0.5}{0.5 + \tau}\right) \overset{\text{def}}{=} r_{j,\alpha}, \tag{17}
$$

where we have used the definition of the function $A_{\alpha,r}$.

Now, the regions where the classifier $\mathrm{Smooth}_\beta(h * p_\alpha)$ predicts 1 are similarly given by $I_{j,\alpha,\beta} = \{x \in I_{j,\alpha} : \psi(h * p_\alpha) * p_\beta \geq 0.5\}$. We can follow the same steps as earlier on the $\alpha$-shrunk balls $I_{j,\alpha} = B_{\ell_2}(c_j, r_{j,\alpha})$. This time, for $x \in I_{j,\alpha}$ we have

$$
\begin{aligned}
\mathrm{Smooth}_\beta(\psi(h * p_\alpha))(x) &= \int_{I_{j,\alpha}} \psi(h * p_\alpha)(t)p_\beta(x - t)dt + \int_{\mathcal{X} \setminus I_{j,\alpha}} \psi(h * p_\alpha)(t)p_\beta(x - t)dt \\
&= \int_{I_{j,\alpha}} 1 \cdot p_\beta(x - t)dt = \Phi_\beta(x - c_j, r_{j,\alpha}).
\end{aligned}
$$

$\beta-$**Shrinkage**   Again, we can find the $\beta$-shrinkage in $I_{j,\alpha}$ as

$$
\Phi_\beta(x - c_j, r_{j,\alpha}) \geq 0.5 \implies \|x - c_j\|_2 \leq A_{\alpha,r_{j,\alpha}}(0.5) \overset{\text{def}}{=} r_{j,\alpha,\beta}.
$$

We have thus obtained $I_{j,\alpha,\beta}$ as the $\alpha,\beta$-shrunk balls, i.e., $I_{j,\alpha,\beta} = B_{\ell_2}(c_j, r_{j,\alpha,\beta})$. We can now compute the risk of the smoothed classifier as

$$R(\text{Smooth}_\beta(\psi(h * p_\alpha)))$$
$$= \int_{\mathcal{Y}} \int_{\mathcal{X}} |\text{Smooth}_\beta(\psi(h * p_\alpha))(x) - Y| p(x, y) dx dy$$
$$= \int_{\mathcal{X}} |\text{Smooth}_\beta(\psi(h * p_\alpha))(x) - 1| p(1|x) p_X(x) dx + \int_{\mathcal{X}} |\text{Smooth}_\beta(\psi(h * p_\alpha)(x)| p(0|x) p_X(x) dx.$$

Let $\mathcal{I} = I_1 \cup I_2 \ldots \cup I_k$ be the positive regions, and $\mathcal{I}_{\alpha,\beta} = I_{1,\alpha,\beta} \cup I_{2,\alpha,\beta} \ldots \cup I_{k,\alpha,\beta}$ be the shrunk positive regions. For the first term above, we note that

$$|\text{Smooth}_\beta(\psi(h * p_\alpha))(x) - 1| p(1|x) = \begin{cases} 0, & x \in \mathcal{I}_{\alpha,\beta} \\ 0.5 + \tau, & x \in \mathcal{I} \setminus \mathcal{I}_{\alpha,\beta} \\ 0, & x \in \mathcal{X} \setminus \mathcal{I} \end{cases}.$$

Similarly, for the second term, we see that

$$\text{Smooth}_\beta(\psi(h * p_\alpha))(x) p(0|x) = \begin{cases} 0.5 - \tau, & x \in \mathcal{I}_{\alpha,\beta} \\ 0, & x \in \mathcal{I} \setminus \mathcal{I}_{\alpha,\beta} \\ 0, & x \in \mathcal{X} \setminus \mathcal{I} \end{cases}.$$

Substituting into the integrals, we obtain the risk of the smoothed classifier

$$R(\text{Smooth}_\beta(\psi(h * p_\alpha))) = (0.5 + \tau) p_X(\mathcal{I} \setminus \mathcal{I}_{\alpha,\beta}) + (0.5 - \tau) p_X(\mathcal{I}_{\alpha,\beta})$$
$$= (0.5 + \tau) p_X(\mathcal{I}) - 2\tau p_X(\mathcal{I}_{\alpha,\beta}).$$

Recalling that $p_X(\cdot)$ is positive everywhere in the domain, and using $\mathcal{I}_{0,\beta} \supset \mathcal{I}_{\alpha,\beta}$ for $0 < \alpha, \beta < \underline{\varsigma}_h$ in the above, we obtain

$$p_X(\mathcal{I}_{0,\beta}) > p_X(\mathcal{I}_{\alpha,\beta}) \implies -2\tau p_X(\mathcal{I}_{\alpha,\beta}) > -2\tau p_X(\mathcal{I}_{0,\beta}).$$

In other words, we have

$$R(\text{Smooth}_\beta(\psi(h * p_\alpha))) > R(\text{Smooth}_\beta(\psi(h))) \implies \Delta_{\alpha,\beta} > \Delta_{0,\beta}. \quad \square$$

## E  Proof of Theorem 4.2

**Theorem 4.2.** *For nice noise distributions $p_\alpha, p_\beta$, for any $h$ supported on a bounded domain $\mathcal{X}$,*

$$\Delta_{\alpha,\beta}(h) \leq 1 - \sum_k p_X(B_{\ell_2}(\hat{x}^k, (\omega_{h,\tau}^k - r_{\alpha,\beta}^k)_+)), \tag{3}$$

*where $r_\alpha^k = \sqrt{\Psi_\alpha^{-1}\left(\frac{0.5}{0.5 + \tau}\right)}$ if $\mathcal{I}_k$ is a positive partition, i.e., $\mathcal{I}_k \in \mathcal{I}^+$ and $r_\alpha^k = \sqrt{\Psi_\alpha^{-1}\left(\frac{0.5}{0.5 - \tau}\right)}$ otherwise, i.e., $\mathcal{I}_k \in \mathcal{I}^-$. Further, $r_{\alpha,\beta}^k = r_\alpha^k + \sqrt{\Psi_\beta^{-1}(0.5)}$, and $\Psi_\alpha, \Psi_\beta$ are the CDFs of the distribution of $\|z\|_2^2$ when $z \sim p_\alpha$ and $z \sim p_\beta$, respectively. $p_X(S)$ denotes the measure of the set $S$ under $p_X$. The upper bound (3) holds for any choice of $\{(\omega_{h,\tau}^k, \hat{x}^k)\}$ such that $\hat{x}^k$ is the center of a ball of radius $\omega_{h,\tau}^k$ completely contained in $\mathcal{I}_{k,\tau}$ defined as $\mathcal{I}_{k,\tau} = \{x \in \mathcal{I}_k : |h(x) - 0.5| \geq \tau\}$. We can choose the sequence $\{(\omega_{h,\tau}^k, \hat{x}^k)\}$ such that the upper bound (3) is minimized.*

**Definition E.1** (Inradius). *The inradius of a set $S$ is defined as the largest radius $r^*$ such that an $\ell_2$ ball of radius $r^*$ is completely contained in $S$. In other words, there exists $x \in S$ such that $B_{\ell_2}(x, r^*) \subseteq S$.*

In order to bound $\Delta_{\alpha,\beta}(h)$, we will follow a strategy which generalizes the arguments in Theorem 4.1 to any arbitrary conditional distribution $h(x) = p(Y = 1|X = x)$. Our first step would be to upper-bound the positive partitions of $\psi(h * p_\alpha)$, i.e., the regions where the classifier trained with noise-augmentation predicts 1. This will be then be used along with properties of randomized smoothing to upper-bound the positive partitions of $\text{Smooth}_\beta(\psi(h * p_\alpha))$, i.e., the regions where the final smoothed classifier predicts 1. These regions would finally be used to upper bound the risk of the smoothed classifier as a function of $\alpha$ and $\beta$.

*Proof.* Let $\mathcal{I}^+$ denote the set of maximal, non-empty simply connected subsets $I \subseteq \mathcal{X}$ of the input space such that $h(x) \geq 0.5$ for each $x \in I$. Pick any $I \in \mathcal{I}^+$ and let $I_\tau \subseteq I$ be such that $h(x) \geq 0.5 + \tau$ for each $x \in I_\tau$ and $0 \leq \tau < 0.5$.

**Bounding region where $\psi(h * p_\alpha)(x) = 1$.** Let $J_1 \subseteq I_\tau$ be the maximal subset of $I_\tau$ such that the noise-trained classifier $\psi(h * p_\alpha)$ predicts 1 in $J_1$, i.e. $J_1 = \{x \in I_\tau : C_1(x) \text{ is satisfied }\}$, where $C_1(x)$ is the condition $(h * p_\alpha)(x) \geq 0.5$. We will firstly lower bound the area in $J_1$ by using a stronger condition $C_2(x) \equiv \int_{I_\tau-x}(\tau + 0.5)p_\alpha(\delta)d\delta \geq 0.5$. To show that $C_2$ is a stronger condition than $C_1$, we will show that $C_2(x) = 1$ implies $C_1(x) = 1$, as follows:

$$(h * p_\alpha)(x) = \int_{\mathcal{X}} h(x - \delta)p_\alpha(\delta)d\delta$$

$$\geq \int_{I_\tau} h(\delta)p_\alpha(x - \delta)d\delta$$

$$\geq \int_{I_\tau} (\tau + 0.5)p_\alpha(x - \delta)d\delta = \int_{I_\tau-x} (\tau + 0.5)p_\alpha(\delta)d\delta$$

Defining $J_2 = \{x \in I_\tau : C_2(x) = 1\}$, we see that $J_2 \subseteq J_1$. We will further lower bound $J_2$ by considering the largest $d$-dimensional ball centered at 0 that can be inscribed in $I_\tau - x$ for each $x$. Define the condition $C_3(x, r)$ as the following:

$$C_3(x, r) \equiv (C_2(x) = 1 \text{ and } B(x, r) \subseteq I_\tau)$$

$$\equiv \left( \underbrace{\int_{I_\tau-x} (\tau + 0.5)p_\alpha(\delta)d\delta \geq 0.5}_{\text{I}} \text{ and } \underbrace{B(x, r) \subseteq I_\tau}_{\text{II}} \right) \tag{18}$$

The set $J_3(r) = \{x \in I_\tau : C_3(x, r) = 1\}$ is a subset of $J_2$ for all $r > 0$. In words, $J_3(r)$ denotes the set of points in $J_2$ which are at least $r$ distance away from the boundary of $I_\tau$. Finally, we combine the two conditions I and II in $C_3(x, r)$ to obtain our final condition $C_4(x, r)$.

$$C_4(x, r) \equiv \int_{B(0,r)} (\tau + 0.5)p_\alpha(\delta)d\delta \geq 0.5 \text{ and } B(x, r) \subseteq I_\tau \tag{19}$$

Note that $B(x, r) \subseteq I_\tau$ is the same as $B(0, r) \subseteq I_\tau - x$. Hence, the integral in (19) is over a smaller set than in (18), from where it follows that $C_4$ is a stronger condition than $C_3$ and the set $J_4(r) = \{x \in I_\tau : C_4(x, r) = 1\}$ is a subset of $J_3(r)$ for all $r > 0$.

We now simplify $C_4(r)$ by using the distribution of $\|z\|_2^2$ where $z \sim p_\alpha$, as follows.

$$\int_{B(0,r)} p_\alpha(\delta)d\delta = \Pr_{z \sim p_\alpha}[\|z\|_2^2 \leq r^2]$$

$$= \Psi_\alpha(r^2)$$

In the above, $\Psi_\alpha$ is the CDF of the distribution of $\|z\|_2^2$ when $z \sim p_\alpha$. Setting $(\tau + 0.5)\Psi_\alpha(r^2) \geq 0.5$ gives us the following condition

$$
\begin{aligned}
C_5(x) &\equiv \left[ B(x, r) \subseteq I_\tau \text{ such that } r \geq \sqrt{\Psi_\alpha^{-1}\left(\frac{0.5}{0.5 + \tau}\right)} \right] \\
&\equiv \left[ B\left(x, \sqrt{\Psi_\alpha^{-1}\left(\frac{0.5}{0.5 + \tau}\right)}\right) \subseteq I_\tau \right]
\end{aligned}
$$

Our final subset $J_5(\alpha) = \{x \in I_\tau : C_5(x) = 1\}$ denotes the set of points in $I$ which are at least $r_\alpha = \sqrt{\Psi_\alpha^{-1}\left(\frac{0.5}{0.5+\tau}\right)}$ away from the boundary of $I_\tau$. In other words, $J_5(\alpha) \subseteq I_\tau$ is a $r_\alpha$-contraction of $I_\tau$.

**Bound region where $\text{Smooth}_\beta(\psi(h * p_\alpha)) = 1$.** We will now follow the same general strategy as in the previous part of the proof. Let $K_1 \subseteq J_5(\alpha)$ be the maximal subset of $J_5(\alpha)$ such that the smoothed classifier $\text{Smooth}_\beta(\psi(h * p_\alpha)$ predicts 1 in $K_1$, i.e., $K_1 = \{x \in J_5(\alpha) : D_1(x) \text{ is satisfied }\}$, where $D_1(x)$ is the condition $(\text{Smooth}_\beta(\psi(h * p_\alpha)))(x) \geq 0.5$. We will firstly lower bound the area in $K_1$ by using a stronger condition $D_2(x) \equiv \int_{J_5(\alpha)-x} p_\beta(\delta)d\delta \geq 0.5$. To show that $D_2$ is a stronger condition than $D_1$, we will show that $D_2(x) = 1$ implies $D_1(x) = 1$, as follows:

$$
\begin{aligned}
(\text{Smooth}_\beta(\psi(h * p_\alpha)))(x) &= \int_{\mathcal{X}} (\psi(h * p_\alpha))(x - \delta)p_\beta(\delta)d\delta \\
&\geq \int_{J_5(\alpha)} (\psi(h * p_\alpha))(\delta)p_\beta(x - \delta)d\delta \\
&= \int_{J_5(\alpha)} p_\beta(x - \delta)d\delta = \int_{J_5(\alpha)-x} p_\beta(\delta)d\delta
\end{aligned}
$$

Defining $K_2 = \{x \in J_5(\alpha) : D_2(x) = 1\}$, we see that $K_2 \subseteq K_1$. We will further lower bound $K_2$ by considering the largest $d$-dimensional ball centered at 0 that can be inscribed in $J_5(\alpha) - x$ for each $x$. Define the condition $D_3(x, r)$ as the following:

$$
\begin{aligned}
D_3(x, r) &\equiv (D_2(x) = 1 \text{ and } B(x, r) \subseteq J_5(\alpha)) \\
&\equiv \left( \int_{J_5(\alpha)-x} p_\beta(\delta)d\delta \geq 0.5 \text{ and } B(x, r) \subseteq J_5(\alpha) \right)
\end{aligned}
\tag{20}
$$

The set $K_3(r) = \{x \in J_5(\alpha) : D_3(x, r) = 1\}$ is a subset of $K_2$ for all $r > 0$. In words, $K_3(r)$ denotes the set of points in $K_2$ which are atleast $r$ distance away from the boundary of $J_5(\alpha)$. Finally, we combine the two conditions in $D_3(x, r)$ to obtain our final condition $D_4(x, r)$.

$$
D_4(x, r) \equiv \int_{B(0,r)} p_\beta(\delta)d\delta \geq 0.5 \text{ and } B(x, r) \subseteq J_5(\alpha)
\tag{21}
$$

Note that $B(x, r) \subseteq J_5(\alpha)$ is the same as $B(0, r) \subseteq J_5(\alpha) - x$. Hence, the integral in (21) is over a smaller set than in (20), from where it follows that $D_4$ is a stronger condition than $D_3$ and the set $K_4(r) = \{x \in J_5(\alpha) : D_4(x, r) = 1\}$ is a subset of $K_3(r)$ for all $r > 0$.

We now simplify $K_4(r)$ by using the distribution of $\|z\|_2^2$ where $z \sim p_\beta$, as follows.

$$
\begin{aligned}
\int_{B(0,r)} p_\beta(\delta)d\delta &= \Pr[\|z\|_2^2 \leq r^2] \text{ where } z \sim p_\beta \\
&= \Psi_\beta(r^2)
\end{aligned}
$$

In the above, $\Psi_\beta$ is the CDF of the distribution of $\|z\|_2^2$ when $z \sim p_\beta$. Setting $\Psi_\beta(r^2) \geq 0.5$ gives us the following condition

$$D_5(x) \equiv \left[ B(x,r) \subseteq J_5(\alpha) \text{ such that } r \geq \sqrt{\Psi_\beta^{-1}(0.5)} \right]$$

$$\equiv \left[ B\left(x, \sqrt{\Psi_\beta^{-1}(0.5)}\right) \subseteq J_5(\alpha) \right]$$

Our final subset $K_5(\alpha, \beta) = \{x \in I_\tau \colon D_5(x) = 1\}$ denotes the set of points in $I$ which are at least $r_{\alpha,\beta} = r_\alpha + \sqrt{\Psi_\beta^{-1}(0.5)}$ away from the boundary of $I_\tau$. In other words, $K_5(\alpha, \beta) \subseteq I_\tau$ is a $r_{\alpha,\beta}$-contraction of $I_\tau$.

We recall that $h$ is such that the subset $I_\tau$ has inradius $\omega_{h,\tau}$. By the definition of inradius, this means that there exists a ball $B(\hat{x}, \omega_{h,\tau})$ for some $\hat{x} \in I_\tau$ such that $B(\hat{x}, \omega_{h,\tau}) \subseteq I_\tau$. Hence, the contraction of $I_\tau$ by $r_{\alpha,\beta}$, i.e., $K_5(\alpha, \beta)$, is a superset of the contraction of $B(\hat{x}, \omega_{h,\tau})$ by $r_{\alpha,\beta}$,

$$B(\hat{x}, \omega_{h,\tau} - r_{\alpha,\beta}) \subseteq K_5(\alpha, \beta). \tag{$\bullet$}$$

**Risk Upper Bound** Recall that $X \sim p_X, Y \sim p_Y$ are the data and the response variables respectively. Let $S$ be the random variable obtained from the smoothed, noise-augmented classifier as $S = \text{Smooth}_\beta(\psi(h * p_\alpha))(X)$. Let $T$ be the random variable obtained from the original classifier $T = \psi(h)(X)$. We know that the risk of the original classifier is given by

$$R(\psi(h)) = \Pr[\psi(h)(X) \neq Y] = p(T = 1, Y = 0) + p(T = 0, Y = 1).$$

Define $\mathcal{I}^+ = \{x \colon \psi(h)(x) = 1\}$ and $\mathcal{I}^- = \{x \colon \psi(h)(x) = 0\}$ as the regions of the input space classified as 1 and 0 respectively by the base classifier. With this definition, we can rewrite the base risk as

$$R(\psi(h)) = p(X \in \mathcal{I}^+, Y = 0) + p(X \in \mathcal{I}^-, Y = 1).$$

Let $\mathcal{K}^+ \subseteq \mathcal{I}^+$ be any subset classified as 1 by the smoothed classifier, i.e.,

$$\mathcal{K}^+ \subseteq \{x \colon \text{Smooth}_\beta(\psi(h * p_\alpha))(x) = 1\} \cap \mathcal{I}^+.$$

Similarly,

$$\mathcal{K}^- \subseteq \{x \colon \text{Smooth}_\beta(\psi(h * p_\alpha))(x) = 0\} \cap \mathcal{I}^-.$$

We will think of the sets $\mathcal{K}$ as shrunk versions of $\mathcal{I}$, and where the smoothed classifier predicts the correct label. Specifically, we will let $\mathcal{K}^+$ be the union of all the contractions $K_5(\alpha, \beta)$ of the positive partitions of $\psi(h)$, where $K_5(\alpha, \beta)$ was obtained earlier. Similarly, we will let $\mathcal{K}^-$ be the union of the contractions of all the negative partitions of $\psi(h)$. This ensures that

$$\bar{\mathcal{K}} \stackrel{\text{def}}{=} \mathcal{X} \setminus (\mathcal{K}^+ \cup \mathcal{K}^-) = \mathcal{X} \setminus \bigcup_k K_{5,k}(\alpha, \beta),$$

where we define $\bar{\mathcal{K}}$ as the rest of the space.

The risk of the smoothed classifier is

$$R(\text{Smooth}_\beta(\psi(h * p_\alpha))) = p(S = 1, Y = 0) + p(S = 0, Y = 1),$$

where the first term decomposes into

$$p(S = 1, Y = 0) = p(S = 1, Y = 0, X \in \mathcal{K}^+) + p(S = 1, Y = 0, X \in \bar{\mathcal{K}})$$
$$= p(Y = 0, X \in \mathcal{K}^+) + p(S = 1, Y = 0, X \in \bar{\mathcal{K}}),$$

noting that $p(S = 1, Y = 0, X \in \mathcal{K}^-) = 0$, as $X \in \mathcal{K}^-$ implies $S = 0$. Similarly, the second term decomposes as

$$p(S = 0, Y = 1) = p(Y = 1, X \in \mathcal{K}^-) + p(S = 0, Y = 1, X \in \bar{\mathcal{K}}).$$

Combining, we have

$$R(\text{Smooth}_\beta(\psi(h * p_\alpha))) = p(Y = 0, X \in \mathcal{K}^+) + p(Y = 1, X \in \mathcal{K}^-) +$$
$$p(S = 1, Y = 0, X \in \bar{\mathcal{K}}) + p(S = 0, Y = 1, X \in \bar{\mathcal{K}})$$
$$= p(Y = 0, X \in \mathcal{K}^+) + p(Y = 1, X \in \mathcal{K}^-) + p(S \neq Y, X \in \bar{\mathcal{K}}).$$

The above expression says that the errors made by the smoothed classifier can be decomposed into the errors made in the regions $\mathcal{K}$, and the error everywhere else $\bar{\mathcal{K}}$. Now, since $\mathcal{K}^+ \subseteq \mathcal{I}^+, \mathcal{K}^- \subseteq \mathcal{I}^-$, we have

$$R(\text{Smooth}_\beta(\psi(h * p_\alpha))) \leq p(Y = 0, X \in \mathcal{I}^+) + p(Y = 1, X \in \mathcal{I}^-) + p(S \neq Y, X \in \bar{\mathcal{K}})$$
$$\leq R(\psi(h)) + p(S \neq Y, X \in \bar{\mathcal{K}})$$
$$\implies R(\text{Smooth}_\beta(\psi(h * p_\alpha))) \leq R(\psi(h)) + p(X \in \bar{\mathcal{K}}) \implies \Delta_{\alpha,\beta}(h) \leq p(X \in \bar{\mathcal{K}}) \qquad (*)$$

As $K_5(\alpha, \beta) \subseteq K_5(0, 0)$ for any $\alpha, \beta \geq 0$, we have $p(X \in \bar{\mathcal{K}}_{\alpha,\beta}) \geq p(X \in \bar{\mathcal{K}}_{0,0})$, showing that the upper-bound $(*)$ decreases as $\alpha$ increases at any fixed $\beta$.

Finally, notice that the upper-bound $(*)$ becomes tight at $\alpha = \beta = 0$, $\mathcal{K}^+ = \mathcal{I}^+, \mathcal{K}^- = \mathcal{I}^-, \bar{\mathcal{K}} = \{\}$, as then $R(\text{Smooth}_0(\psi(h * p_0))) = R(\psi(h))$ and $p(X \in \bar{\mathcal{K}}) = 0$.

Finally, we can use ($\bullet$) in $(*)$ to obtain

$$\Delta_{\alpha,\beta} \leq 1 - \sum_k p_X(B_{\ell_2}(\hat{x}^k, \omega_{h,\tau}^k - r_{\alpha,\beta}^k)),$$

for any choice of the points $\{\hat{x}_k\}$ such that $\hat{x}^k$ is such that $B_{\ell_2}(\hat{x}^k, \omega_{h,\tau}^k) \subseteq I_{k,\tau}$.

Finally, the quantities $\Psi_\alpha^{-1}\left(\frac{0.5}{0.5 \pm \tau}\right)$ and $\Psi_\beta^{-1}(0.5)$ must be non-negative for the above balls to be well defined. This in turn implies that $\omega_{h,\tau}^k - r_{\alpha,\beta}^k \geq 0$ for all $k$, i.e., the inradius of each partition is at least as large as the amount of shrinkage that partition goes through. These requirements are *not* additional constraints on $h$, but rather a property of the proof technique: whenever the inradius is too small, the partition vanishes from the upper bound. To capture this subtlety, we can write the upper bound as

$$\Delta_{\alpha,\beta} \leq 1 - \sum_k p_X(B_{\ell_2}(\hat{x}^k, (\omega_{h,\tau}^k - r_{\alpha,\beta}^k)_+)),$$

where $(c)_+$ denotes the positive part of $c$. $\qquad \square$

## F   Proof of Theorem 4.3

**Theorem 4.3.** *There exist nice distributions $p_\alpha, p_\beta$, a family $\mathcal{H}_2$ with low interference distance $\overline{\zeta_{\mathcal{H}_2}}$, and data-distributions $p_X$, such that for all $h \in \mathcal{H}_2$ we have $\Delta_{0,\beta_0}(h) > \Delta_{\alpha,\beta_0}(h)$ for some $\alpha > 0, \beta_0 > 0$.*

*Proof.* We will firstly derive general conditions on $h \in \mathcal{H}_2$ and $\alpha_0, \beta_0 > 0$ such that $\Delta_{\alpha_0,\beta_0}(h) < \Delta_{0,\beta_0}(h)$. We will then show that the $\mathcal{H}_2$ as restricted by these conditions is not empty, by providing a particular $h$ that satisfies these conditions. We will assume that the domain is bounded.

**Lower-Bound on $\beta$.**   We first of all require $\beta$ to be large enough such that $\psi(\psi(h) * p_{\beta_0})(x) = 0$ for all $x \in \mathcal{X}$. This is to ensure that without noise-augmentation, the smoothed classifier predicts 0 everywhere. In other words, $\forall x \in \mathcal{X}$ we require

$$\int_{t \in \mathcal{X}} \psi(h)(x - t) p_{\beta_0}(t) dt \leq 0.5. \qquad (22)$$

Condition (22) implies $\beta_0 \geq \underline{\beta}$ for some $\underline{\beta}$, as will become clear later in (26).

**Acceptable Range of $\alpha$.** Secondly, we require $\psi(h * p_{\alpha_0})(x) = 1$ over a set $\bar{\mathcal{X}}$, where we think of the size of $\bar{\mathcal{X}}$ as being *close* to the size of $\mathcal{X}$ in the sense $p_X(x \in \mathcal{X} \setminus \bar{\mathcal{X}}) \leq \epsilon$ for a small $\epsilon$. This subset $\bar{\mathcal{X}}$ denotes the part of the input where the noise-trained classifier predicts 1. In other words, for all $x \in \bar{\mathcal{X}}$, we require

$$\int_{t \in \mathcal{X}} h(x - t) p_{\alpha_0}(t) dt \geq 0.5. \tag{23}$$

Condition (23) implies lower and upper bounds $\bar{\alpha} \geq \alpha_0 \geq \underline{\alpha}$, as will become clear later in (27), (28).

**Upper-Bound on $\beta$.** Thirdly, we require $\psi(\psi(h * p_{\alpha_0}) * p_{\beta_0})(x) = 1$ for all $x \in \hat{\mathcal{X}}$, where we again think of the size of $\hat{\mathcal{X}}$ as being *close* to the size of $\bar{\mathcal{X}}$, i.e., $p_X(\bar{\mathcal{X}} \setminus \hat{\mathcal{X}}) \leq \epsilon$. This ensures that the smoothed classifier predicts 1 in almost all of the region where the noise-trained classifier predicts 1. In other words, for all $x \in \hat{\mathcal{X}}$, we require

$$\int_{t \in \mathcal{X}} \psi(h * p_{\alpha_0})(x - t) p_{\beta_0}(t) dt \geq 0.5. \tag{24}$$

Condition (24) implies an upper bound $\bar{\beta} \geq \beta_0$, as will become clear in (29).

**Small Interference-Distance.** Finally, we require that the data-distribution places more mass on the label 1, i.e.

$$p(Y = 1) > \frac{1}{2} + \epsilon. \tag{25}$$

Condition (25) ensures that $h$ has a small upper-interference distance. Informally, this is expected: (25) says that the positive partitions of $\psi(h)$ need to have a certain mass under $p$, and the domain is bounded, so the partitions cannot be far apart. Formally, this will become clear in (30) when we demonstrate a particular example $h$ where the above constraints are satisfied. Before that, we will prove that the above conditions are sufficient to guarantee $\Delta_{\alpha_0, \beta_0}(h) < \Delta_{0, \beta_0}(h)$.

Let $S_{\alpha, \beta}$ be the random variable $\text{Smooth}_\beta(\psi(h * p_\alpha))(X), X \sim p_X$, and simplify $\Delta_{0, \beta_0}$ as

$$\begin{aligned}
\Delta_{0, \beta_0}(h) &= p(Y \neq S_{0, \beta_0}) & \text{Using (22)} \\
&= p(Y \neq 0) \\
&> p(Y = 0) + 2\epsilon & \text{Using (25)} \\
&\geq p(Y = 0, X \in \hat{\mathcal{X}}) + \epsilon + \epsilon \\
&\geq p(Y \neq 1, X \in \hat{\mathcal{X}}) + p_X(\bar{\mathcal{X}} \setminus \hat{\mathcal{X}}) + p_X(\mathcal{X} \setminus \bar{\mathcal{X}}) \\
&= p(Y \neq S_{\alpha_0, \beta_0}, X \in \hat{\mathcal{X}}) + p(X \in \mathcal{X} \setminus \hat{\mathcal{X}}) & \text{Using (24)} \\
&\geq p(Y \neq S_{\alpha_0, \beta_0}, X \in \hat{\mathcal{X}}) + p(Y \neq S_{\alpha_0, \beta_0}, X \in \mathcal{X} \setminus \hat{\mathcal{X}}) \\
&= \Delta_{\alpha_0, \beta_0}(h).
\end{aligned}$$

We will consider the one-dimensional example $\bar{h}$ described in Section 4.2, and show that it satisfies the above constraints. The domain can be taken to be $\mathcal{X} = [-0.25, 0.25]$. $\bar{h}$ is defined by specifying $c_1, c_2, c_3, c_4$. We can take $c_1 = -0.25$, $c_2 = -0.25 + \omega$, $c_4 = 0.25$, $c_3 = 0.25 - \omega$, for some $\omega \leq 0.25$. Here $\omega$ is the inradius of the positive partitions of $\bar{h}$, similar to what we saw in Theorem 4.2. Additionally, the upper interference distance $\bar{\zeta}$ is the distance between the positive partitions, i.e., $\bar{\zeta} = (0.25 - \omega) - (-0.25 + \omega) = 0.5 - 2\omega$. The smoothing distribution $p$ is uniform in a ball, as $p_\theta = \text{Unif}([-\theta/2, \theta/2])$, and the marginal distribution $p_X$ is uniform over $[-0.25, 0.25]$.

The lower bound on $\beta$ (22) then requires for all $x \in \mathcal{X}$, we have $\int_{-\beta/2}^{\beta/2} \psi(h)(x - t) \cdot (1/\beta) dt \leq 0.5$. A stronger condition is $(1/\beta) \int_{-0.25}^{0.25} \psi(h)(t) dt \leq 0.5$ which implies a lower bound on $\beta$ as

$$(1/\beta)(2\omega) \leq 0.5 \implies \beta \geq 4\omega. \tag{26}$$

The acceptable range of $\alpha$ is defined by (23). We take $\bar{\mathcal{X}} = \mathcal{X}$ (hence $\epsilon = 0$). For $x \in [-0.25 + \omega, 0.25 - \omega]$, (23) gives

$$2 \cdot (0.25 - \omega)(1/\alpha) \leq 0.5 \implies \alpha \geq 4(0.25 - \omega). \tag{27}$$

For $x \in [-0.25, -0.25 + \omega] \cup [0.25 - \omega, 0.25]$, a stronger condition to (23) gives

$$\omega(1/\alpha) \geq 0.5 \implies \alpha \leq 2\omega. \tag{28}$$

The upper bound on $\beta$ is given by (24). As $\epsilon = 0$, we have $\hat{\mathcal{X}} = \bar{\mathcal{X}}$, which gives

$$(0.5) \cdot (1/\beta) \geq 0.5 \implies \beta \leq 1.. \tag{29}$$

Finally, Eq. (25) gives a condition on $\omega$:

$$(2\omega) \cdot \frac{1}{0.5} > \frac{1}{2} \implies \omega > 0.125. \tag{30}$$

The proof is now complete with the observation that $\omega = 0.23, \alpha = 0.1, \beta = 0.93$ satisfies these constraints, showing that $\bar{h} \in \mathcal{H}_2$ as $\Delta_{0,0.93}(\bar{h}) > \Delta_{0.1,0.93}(\bar{h})$.

Note that the condition (30) on $\omega$ implied an upper-bound on the interference distance $\bar{\zeta}$, as $\bar{\zeta} = 0.5 - 2\omega$. This shows that when $h$ satisfies the condition (25), $h$ has a low interference distance. $\qquad\square$

## G    Accommodating Errors in Learning Bayes Classifiers.

While obtaining a randomized-smoothed classifier in a real learning scenario, we might deviate from the Bayes classifier in any stage of the process. Specifically, we have 2 stages: (1) Learn a classifier on the noisy data $(X_s, Y)$ and (2) Smooth the resultant classifier using randomized smoothing.

In Theorem 4.1, we assumed that we are able to exactly learn the Bayes Classifier for $(X_s, Y)$ in stage (1), i.e. $\psi(h * p_\alpha)$. This assumption can be relaxed while maintaining the same general proof technique, albeit by paying with further slack in the obtained bounds due to the inexactness of the learned classifier. We now sketch a modification of the argument in Appendix D that allows a bounded deviation from the Bayes classifier.

We start with the first column of Figure 1, i.e. $\psi(h)$, and train the classifier on data perturbed by noise $p_\alpha$. In the ideal case sketched above, this leads precisely to the second column, i.e., $\psi(h * p_\alpha)$. In the non-ideal scenario, we assume that we obtain a classifier $g$ in stage (1) after training on $(X_s, Y)$, such that for all $x$, we have $|g(x) - h * p_\alpha(x)| \leq \eta$. In other words, the maximum error in the trained classifier compared to the actual conditional is no larger than $\eta$. For clarity, we still maintain that stage (2) is done exactly, i.e. that given $\psi(g)$ we are able to obtain $\text{Smooth}_\beta(\psi(g))$ exactly as our final randomized-smoothed classifier.

As a result, we are interested in analysing the modified excess risk

$$\Delta_{\alpha,\beta}(g, h) = R(\text{Smooth}_\beta(\psi(g))) - R(\psi(h)), \tag{31}$$

for all $(g, h)$ such that $\|g - h * p_\alpha\|_\infty \leq \eta$. Eq. (31) captures the increase in the benign risk due to randomized smoothing an imperfectly learned classifier on the noise-augmented data. Additionally, we define

$$\overline{\Delta_{\alpha,\beta}^\eta}(h) = \max_{\{g \,:\, \|g - h * p_\alpha\|_\infty \leq \eta\}} \Delta_{\alpha,\beta}(g, h), \tag{32}$$

to be the maximum excess risk when using an imperfectly learned $g$. Similarly, we define

$$\underline{\Delta_{\alpha,\beta}^\eta}(h) = \min_{\{g \,:\, \|g - h * p_\alpha\|_\infty \leq \eta\}} \Delta_{\alpha,\beta}(g, h), \tag{33}$$

to be the minimum excess risk when using an imperfectly learned $g$. We now have the following modified version of Theorem 4.1.

**Theorem G.1.** *For nice noise distributions $p_\alpha = \text{Unif}(B_{\ell_2}(0, \alpha))$ and $p_\beta = \text{Unif}(B_{\ell_2}(0, \beta))$, there exists $\mathcal{H}_1$ with interference distance $\underline{\zeta}_{\mathcal{H}_1}$, such that for all $h \in \mathcal{H}_1$ we have $\overline{\Delta_{0,\beta}^\eta}(h) < \overline{\Delta_{\alpha,\beta}^\eta}(h)$, and $\underline{\Delta_{0,\beta}^\eta}(h) < \underline{\Delta_{\alpha,\beta}^\eta}(h)$ for all smoothing parameters $\alpha, \beta$ such that $\underline{\zeta}_{\mathcal{H}_1} > 2\max(\alpha, \beta)$, and $\eta < 0.5$.*

*Proof.* Similar to Theorem 4.1, let $I_1, I_2, \ldots, I_k$ be spheres such that $I_j = B_{\ell_2}(c_j, r)$ for some positive radius $r > 0$. Define $h(x) = 0.5 + \tau$ whenever $x \in I_1 \cup I_2 \ldots \cup I_k$, and $0$ otherwise, for some $0 \le \tau < 0.5$.

**Large Interference Distance**  We assume that $\zeta_h > 2\max(\alpha, \beta)$, where note the additional factor 2 in the required lower bound as compared to Theorem 4.1. This stricter condition will become useful when we deal with errors in learning the bayes classifier.

**$\alpha$−Shrinkage**  Recall that we earlier analyzed the region $I_{j,\alpha} = \{x \in I_j \colon h * p_\alpha(x) \ge 0.5\}$. In this new non-ideal setting, it is now useful to consider the regions $I_{j,\alpha}^{+\eta}$ defined as

$$I_{j,\alpha}^{+\eta} = \{x \in I_{j,\alpha} \colon h * p_\alpha(x) \ge 0.5 + \eta\}. \tag{34}$$

The inaccurate classifier $g$ would have $\psi(g)(x) = 1$ for all $x \in \cup_i I_{i,\alpha}^{+\eta}$, as $g(x) \ge h * p_\alpha(x) - \eta \ge 0.5$.

Similarly, the sets $\{I_{j,\alpha}^{-\eta}\}$ can be defined as the set of maximal simply connected regions forming a partition of $\{x \colon h * p_\alpha(x) \ge 0.5 - \eta\}$. There is a natural correspondence between $I_{j,\alpha}^{-\eta}$, $I_{j,\alpha}^{+\eta}$ and $I_{j,\alpha}$, which will become clear once we obtain the explicit forms of these sets. Note that $\psi(g)(x) = 0$ for all $x \notin \cup_i I_{i,\alpha}^{-\eta}$, as $g(x) \le h * p_\alpha(x) + \eta \le 0.5$.

We can characterize $I_{j,\alpha}^{+\eta}$ explicitly as the set of all $x$ satisfying

$$(0.5 + \tau)\Phi_\alpha(x - c_j, r) \ge 0.5 + \eta \implies \|x - c_j\|_2 \le A_{\alpha,r}\left(\frac{0.5 + \eta}{0.5 + \tau}\right) \overset{\text{def}}{=} r_\alpha^{+\eta}, \tag{35}$$

obtaining the shrunk ball $I_{j,\alpha}^{+\eta} = B(c_j, r_\alpha^{+\eta})$. Similarly, we see that $I_{j,\alpha}^{-\eta}$ is the ball $I_{j,\alpha}^{-\eta} = B(c_j, r_\alpha^{-\eta})$ with radius

$$r_\alpha^{-\eta} \overset{\text{def}}{=} A_{\alpha,r}\left(\frac{0.5 - \eta}{0.5 + \tau}\right).$$

**Handling Inaccuracy in $g$**  We now perform randomized smoothing on this classifier $g$, and obtain $\psi(\psi(g) * p_\beta)$.

For $x \in I_j$ we have that

$$\text{Smooth}_\beta(\psi(g))(x) = \int_{I_{j,\alpha}^{+\eta}} \psi(g)(t)p_\beta(x - t)dt + \int_{I_{j,\alpha}^{-\eta} \setminus I_{j,\alpha}^{+\eta}} \psi(g)(t)p_\beta(x - t)dt + \int_{\mathcal{X} \setminus I_{j,\alpha}^{-\eta}} \psi(g)(t)p_\beta(x - t)dt. \tag{36}$$

We will now split the domain of the last integral in (36) as

$$\mathcal{X} \setminus I_{j,\alpha}^{-\eta} = \left(\cup_{i \neq j} I_{j,\alpha}^{-\eta}\right) \cup \left(\mathcal{X} \setminus \cup_i I_{j,\alpha}^{-\eta}\right),$$

where we note that the integral over the second set in the union, i.e., $\int_{\mathcal{X} \setminus \cup_i I_{i,\alpha}^{-\eta}} \psi(g)(t)p_\beta(x - t)dt$ is zero, by the observation above that $\psi(g)(t) = 0$ for all $t \notin \cup_j I_{j,\alpha}^{-\eta}$. This leaves us with the integral over the first set, which we will now show to be 0 as well.

For the first set, we want to see whether there is any point $t \in I_{i,\alpha}^{-\eta}$, $i \neq j$ such that $\psi(g)(t) = 1$ and $p_\beta(x - t) > 0$. This can happen only when $g(t) \ge 0.5$ and $\|x - t\|_2 \le \beta$. Now consider the triangle formed by the points $\{x, t, c_i\}$, and apply the reverse triangle inequality to get

$$\|t - c_i\|_2 \ge \|c_i - x\|_2 - \|x - t\|_2 \ge 2\max(\alpha, \beta) + r - \beta \ge \alpha + r,$$

where the interference distance condition was used for $\|c_i - x\| > 2\max(\alpha, \beta)$. But now we know that $g(t) - h * p_\alpha(t) \leq \eta < 0.5$, and that $h * p_\alpha(t) = 0$ for any $t$ at a distance more than $\alpha + r$ distance away from all the centers $c_i$. This implies that $g(t) < 0.5$, which further implies that $\psi(g)(t) = 0$. Hence, the integral over the first set $\left(\cup_{i \neq j} I_{j,\alpha}^{-\eta}\right)$ is 0.

The first term in (36) is simply

$$\int_{I_{j,\alpha}^{+\eta}} \psi(g)(t) p_\beta(x - t) dt = \int_{I_{j,\alpha}^{+\eta}} 1 \cdot p_\beta(x - t) dt = \Phi_\beta(x - c_j, r_{j,\alpha}^{+\eta}).$$

For the second term in (36), we can only produce upper and lower bounds, as

$$0 \leq \int_{I_{j,\alpha}^{-\eta} \setminus I_{j,\alpha}^{+\eta}} \psi(g)(t) p_\beta(x - t) dt \leq \int_{I_{j,\alpha}^{-\eta} \setminus I_{j,\alpha}^{+\eta}} 1 \cdot p_\beta(x - t) dt$$

Combining, we have

$$\Phi_\beta(x - c_j, r_\alpha^{+\eta}) \leq \text{Smooth}_\beta(\psi(g))(x) \leq \Phi_\beta(x - c_j, r_\alpha^{-\eta}) \tag{37}$$

where the lower-bound occurs when $g(x)$ takes the lowest possible value for all $x \in I_{j,\alpha}^{+\eta} \setminus I_{j,\alpha}^{-\eta}$. Similarly, the upper-bound occurs when $g(x)$ takes the highest possible value in the same region.

$\beta-$**Shrinkage**   Compared Theorem 4.1, we now only have an inequality (37) for the smoothed classifier at any point $x$. As a result, we cannot determine the positive regions $I_{j,\alpha,\beta} = \{x \in I_j : \text{Smooth}_\beta(\psi(g))(x) \geq 0.5\}$ exactly. Instead, the following set inclusions follow from (37):

$$B_{\ell_2}(c_j, A_{\alpha, r_\alpha^{+\eta}}(0.5)) \subseteq I_{j,\alpha,\beta} \subseteq B_{\ell_2}(c_j, A_{\alpha, r_\alpha^{-\eta}}(0.5)).$$

**Bounding Risk**   Similar to Theorem 4.1, we can now compute the risk of the smoothed classifier as

$$R(\text{Smooth}_\beta(\psi(h * p_\alpha))) \tag{38}$$

$$= \int_{\mathcal{X}} |\text{Smooth}_\beta(\psi(h * p_\alpha))(x) - 1| p(1|x) p_X(x) dx + \int_{\mathcal{X}} \text{Smooth}_\beta(\psi(h * p_\alpha)(x) p(0|x) p_X(x) dx. \tag{39}$$

Let $\mathcal{I} = I_1 \cup I_2 \ldots \cup I_k$ are the positive regions for the base classifier $\psi(h)$, and let $\mathcal{I}_{\alpha,\beta}^{+\eta} = \cup_j B_{\ell_2}(c_j, A_{\alpha, r_\alpha^{+\eta}})$ and $\mathcal{I}_{\alpha,\beta}^{-\eta} = \cup_j B_{\ell_2}(c_j, A_{\alpha, r_\alpha^{-\eta}}(0.5))$ be the upper and lower bounds to the positive regions of the smoothed classifier $\text{Smooth}_\beta(\psi(g))$, i.e., $\mathcal{I}_{\alpha,\beta} = \cup_j I_{j,\alpha,\beta}$.

Let $r_{\alpha,\beta,\eta} \overset{\Delta}{=} A_{\alpha, r_\alpha^{-\eta}}(0.5)$ denote the largest possible radius of the positive partitions after smoothing. We now have two cases, $r_{\alpha,\beta,\eta} \leq r$ (Case A), or $r < r_{\alpha,\beta,\eta}$ (Case B). When $\eta = 0$, the positive partitions always shrink (as we saw in Theorem 3.1), and we are in Case A. As we increase $\eta$ beyond a certain $\eta \geq \eta_0$, we go into Case B, where the positive partitions have potentially dilated after smoothing. We handle both cases separately.

**Case A**   Define $S \subseteq \mathcal{I}_{\alpha,\beta}^{-\eta} \setminus \mathcal{I}_{\alpha,\beta}^{+\eta}$ to be the subset of $\mathcal{I}_{\alpha,\beta}^{-\eta} \setminus \mathcal{I}_{\alpha,\beta}^{+\eta}$ where $\text{Smooth}_\beta(\psi(g))(x) = 1$. For the first integral in (39), we obtain

$$|\text{Smooth}_\beta(\psi(g))(x) - 1| p(1|x) = \begin{cases} 0, & x \in \mathcal{I}_{\alpha,\beta}^{+\eta} \\ 0, & x \in S \\ 0.5 + \tau, & x \in (\mathcal{I}_{\alpha,\beta}^{-\eta} \setminus \mathcal{I}_{\alpha,\beta}^{+\eta}) \setminus S \\ 0.5 + \tau, & x \in \mathcal{I} \setminus \mathcal{I}_{\alpha,\beta}^{-\eta} \\ 0, & x \in \mathcal{X} \setminus \mathcal{I} \end{cases}.$$

For the second integral in (39), we obtain

$$\text{Smooth}_\beta(\psi(g))(x)p(0|x) = \begin{cases} 0.5 - \tau, & x \in \mathcal{I}_{\alpha,\beta}^{+\eta} \\ 0.5 - \tau, & x \in S \\ 0, & x \in (\mathcal{I}_{\alpha,\beta}^{-\eta} \setminus \mathcal{I}_{\alpha,\beta}^{+\eta}) \setminus S \\ 0, & x \in \mathcal{I} \setminus \mathcal{I}_{\alpha,\beta}^{-\eta} \\ 0, & x \in \mathcal{X} \setminus \mathcal{I} \end{cases} .$$

Substituting into the integrals, we obtain

$$R(\text{Smooth}_\beta(\psi(g))) = (0.5 - \tau)p_X(\mathcal{I}_{\alpha,\beta}^{+\eta} \cup S) + (0.5 + \tau)p_X\left( ((\mathcal{I}_{\alpha,\beta}^{-\eta} \setminus \mathcal{I}_{\alpha,\beta}^{+\eta}) \setminus S) \cup (\mathcal{I} \setminus \mathcal{I}_{\alpha,\beta}^{-\eta}) \right),$$

which is minimized when $S$ is as large as possible, and vice-versa. This gives us the risk bounds

$$(0.5 - \tau)p_X(\mathcal{I}_{\alpha,\beta}^{-\eta}) + (0.5 + \tau)p_X(\mathcal{I} \setminus \mathcal{I}_{\alpha,\beta}^{-\eta}) \le R \le (0.5 - \tau)p_X(\mathcal{I}_{\alpha,\beta}^{+\eta}) + (0.5 + \tau)p_X(\mathcal{I} \setminus \mathcal{I}_{\alpha,\beta}^{+\eta})$$

$$\Rightarrow \underbrace{(0.5 + \tau)p_X(\mathcal{I}) - 2\tau p_X(\mathcal{I}_{\alpha,\beta}^{-\eta}) - R(\psi(h))}_{\underline{\Delta}_{\alpha,\beta}^{\eta}(h)} \le \Delta_{\alpha,\beta}(g,h) \le \underbrace{(0.5 + \tau)p_X(\mathcal{I}) - 2\tau p_X(\mathcal{I}_{\alpha,\beta}^{+\eta}) - R(\psi(h))}_{\overline{\Delta}_{\alpha,\beta}^{\eta}(h)} \quad (40)$$

As $\alpha$ increases for a fixed $\beta, \eta, \tau$, both the upper and lower bounds in (40) increase, following the same argument as Theorem 4.1.

**Case B** Define $T_1 \subseteq \mathcal{I}_{\alpha,\beta}^{-\eta} \setminus \mathcal{I}$ to be the subset of $\mathcal{I}_{\alpha,\beta}^{-\eta} \setminus \mathcal{I}$ where $\text{Smooth}_\beta(\psi(g))(x) = 1$. Define $T_2 \subseteq \mathcal{I} \setminus \mathcal{I}_{\alpha,\beta}^{+\eta}$ to be the subset of $\mathcal{I} \setminus \mathcal{I}_{\alpha,\beta}^{+\eta}$ where $\text{Smooth}_\beta(\psi(g))(x) = 1$. For the first integral in (39), we obtain

$$|\text{Smooth}_\beta(\psi(g))(x) - 1|p(1|x) = \begin{cases} 0, & x \in \mathcal{I}_{\alpha,\beta}^{+\eta} \\ 0, & x \in T_2 \\ 0.5 + \tau, & x \in (\mathcal{I} \setminus \mathcal{I}_{\alpha,\beta}^{+\eta}) \setminus T_2 \\ 0, & x \in \mathcal{X} \setminus \mathcal{I} \end{cases} .$$

For the second integral in (39), we obtain

$$\text{Smooth}_\beta(\psi(g))(x)p(0|x) = \begin{cases} 0.5 - \tau, & x \in \mathcal{I}_{\alpha,\beta}^{+\eta} \\ 0.5 - \tau, & x \in T_2 \\ 0, & x \in (\mathcal{I} \setminus \mathcal{I}_{\alpha,\beta}^{+\eta}) \setminus T_2 \\ 1, & x \in T_1 \\ 0, & x \in (\mathcal{I}_{\alpha,\beta}^{-\eta} \setminus \mathcal{I}) \setminus T_1 \\ 0, & x \in \mathcal{X} \setminus \mathcal{I}_{\alpha,\beta}^{-\eta} \end{cases} .$$

Again substituting into the integrals, we obtain

$$R(\text{Smooth}_\beta(\psi(g))) = (0.5 + \tau)p_X((\mathcal{I} \setminus \mathcal{I}_{\alpha,\beta}^{+\eta}) \setminus T_2) + (0.5 - \tau)p_X(T_2) + p_X(T_1) + (0.5 - \tau)p_X(\mathcal{I}_{\alpha,\beta}^{+\eta}),$$

which is minimized when $T_1$ is empty and $T_2$ is as large as possible, and maximized when $T_1$ is as large as possible and $T_2$ is empty. This gives the risk bounds

$$(0.5 - \tau)p_X(\mathcal{I}) \le R(\text{Smooth}_\beta(\psi(g))) \le (0.5 + \tau)p_X(\mathcal{I}) - 2\tau p_X(\mathcal{I}_{\alpha,\beta}^{+\eta}) + p_X(\mathcal{I}_{\alpha,\beta}^{-\eta} \setminus \mathcal{I})$$

$$\Rightarrow \underbrace{(0.5 - \tau)p_X(\mathcal{I}) - R(\psi(h))}_{\underline{\Delta}_{\alpha,\beta}^{\eta}(h)} \le \Delta_{\alpha,\beta}(g,h) \le \underbrace{(0.5 + \tau)p_X(\mathcal{I}) - 2\tau p_X(\mathcal{I}_{\alpha,\beta}^{+\eta}) + p_X(\mathcal{I}_{\alpha,\beta}^{-\eta} \setminus \mathcal{I}) - R(\psi(h))}_{\overline{\Delta}_{\alpha,\beta}^{\eta}(h)} \quad (41)$$

Note that the lower bound for the risk in (41) is equal to the Bayes Error - this is expected as in the best scenario, the classifier after smoothing can be identical to the original classifier in Case B. For the upper bound, we observe that as we increase $\alpha$ at a fixed $\beta, \eta, \tau$, the set $\mathcal{I}_{\alpha,\beta}^{+\eta}$ shrinks, which causes the upper bound to increase.

We have hence shown that in both Case A and Case B, the risk of the smoothed classifier increases as $\alpha$ is increased from 0, modulo approximations due to errors in learning the bayes classifier. $\qquad \square$

**Upper Bound for General $g$**  We show here that via a simple application of Theorem 4.2, we can upper bound $\Delta_{\alpha,\beta}(g,h)$ for general $g, h$:

$$\Delta_{\alpha,\beta}(g,h) \leq \Delta_{\alpha,\beta}(h) + p_X\left((h * p_\alpha)(X) \neq g(X)\right) \tag{42}$$

To show (42), we let $S_g$ denote the random variable $\mathrm{Smooth}_\beta(\psi(g))(X)$, and $S_h$ denote the random variable $\mathrm{Smooth}_\beta(\psi(h) * p_\alpha)(X)$. We use the following simple sequence of upper-bounds:

$$\begin{aligned}
\Delta_{\alpha,\beta}(g,h) &= R(\mathrm{Smooth}_\beta(\psi(g))) - R(\psi(h)) \\
&= p(S_g \neq Y) - R(\psi(h)) \\
&= p(S_g \neq Y, S_h = S_g) - R(\psi(h)) + p(S_g \neq Y, S_h \neq S_g) \\
&\leq p(S_h \neq Y) - R(\psi(h)) + p(S_h \neq S_g) \\
&= \Delta_{\alpha,\beta}(h) + p(S_h \neq S_g) \\
&\leq \Delta_{\alpha,\beta}(h) + p((h * p_\alpha)(X) \neq g(X))
\end{aligned}$$

