# OpenReview forum: "Understanding Noise-Augmented Training for Randomized Smoothing"
_TMLR — Accepted by TMLR_

### Review · Reviewer_UVsX · 2023-02-15

**Summary Of Contributions:**

This paper studies the role of noise augmentation (NA) when training classifiers that are later smoothed during randomized smoothing (RS).  The main idea is to characterize regimes in which NA help/hurt the benign and smoothed performance of a given classifier.  The authors find that the degree to which NA helps depends on a particular metric on the Bayes optimal classifier, which they refer to as the interference distance (ID).  Several experiments are performed to demonstrate the interplay between the ID and the parameters of the distributions used for NA and RS.

**Audience:**

Yes

**Claims And Evidence:**

No

**Requested Changes:**

1. Fixing the grammatical and notational problems outlined above.
2. Having a clearer take home message that is supported **clearly** by the experiments (see the "Weaknesses" section)
3. Resolving the "Miscellaneous" comments above.
4. Defining the ID more clearly.
5. If you're going to use the informal presentation style, make sure the results are stated so that a reader can understand them informally.  I.e., don't include constants or quantities that are completely undefined or under-defined.
6. Move most of the conclusion to the appendix to make room for additional definitions.
7. Add more details to Figures 1 and 2, both of which look great, but are not explained clearly enough.

**Strengths And Weaknesses:**

### Strengths

**Novel problem.**  I am not aware of another paper that looks at the role of NA in RS.  Indeed, NA is widely used for RS, but it isn't clear how one should use NA and why it results in stronger randomized classifiers.  The authors attempt to demystify this phenomenon from a theoretical perspective.  And their theory does lead to (albeit somewhat limited) insights on the practice of NA for RS.

### Weaknesses

**Informal presentation.**  I think that an informal statement of theoretical results can be a useful way of presenting results like this.  However, I don't believe that the presentation is effective here.  To improve this, I would start Section 3 by telling the reader that the results are going to be informally stated.  Then, I would try to make the statements of the results informal.  Right now, they read as half-way between formal and informal.  It's challenging to interpret the theorems in Section 3 without knowing what the ID is.  In Theorem 3.2, at this level of detail, is this not trivial, in the sense that $G_{\alpha,\beta}$ could simply be taken to be $\Delta_{\alpha,\beta}(h)$?  $\Delta$ seems to be monotonic in both of these parameters, right?

**The "simple" example.**  I didn't understand the 1-D example.  It is relatively complicated.  It might help to pick arbitrary constants for $c_1$-$c_4$, hence sacrificing generality for intuitiveness.  Figure 2 needs to be explained in more detail.  You may want to walk the reader through this diagram panel by panel, because as written, it was rather confusing.

**Experiments.**  There is a strange argument at the end of the experiments.  The authors say that

> "The first observation that we make from the MNIST and CIFAR-10 plots in Figure 5 is that a non-zero data-augmentation is always beneficial, i.e., $\alpha^\star > 0$. This suggests that real data distributions lie in the small interference distance regime."

The argument seems to be that since NA is useful, the distribution of data must be in the small ID regime.  However, the results in the paper (i.e., Theorems 3.3 and 4.3) seem to say that small ID implies that NA is useful.  If we let A denote "the distribution has small ID" and we let B denote "NA is useful", this can be simplified as follows: The theory says that $A \implies B$, whereas the experiments say that $B\implies A$.  Clearly, these two are not equivalent.  Perhaps the authors can clarify, because this conflict undermines the main results of the paper.

**Grammatical problems.**  Here's a summary of some of the grammatical issues I found in this paper:
* Page 1: "One of the central questions..." -- the authors do not state an actual question here.
* Page 2: What is meant by "out of the box?"
* Page 3: "...for a pair of input sample and its label" -- this is awkwardly worded
* Page 3: "rendered astray" -- this wording doesn't make sense.  It would improve readability to use more precise language
* Page 4: "relaxed the relax this assumption" -- this seems wrong, consider revising
* Page 4: "Our first result consists in demonstrating..." -- this is not correct grammar

**Notational problems.**
* The symbol $h$ is used in different ways, and they seem to contradict one another.  On page 2, the authors say that $h:\mathcal{X}\to[0,1]$, meaning that $h(x)$ is a number between 0 and 1.  On page 5, $\mathcal{H}_1$ is defined to be a set of distributions, and the authors say that $h\in\mathcal{H}_1$.  So is $h$ a distribution now?  Then, in Section 4, the authors say "the conditional distribution $h(x)$.  $h(x)$ is a number -- how can it be a conditional distribution?

Also, in (2), $\Delta_{\alpha,\beta}$ doesn't have an argument.  In Theorem 3.1, it takes $h$ as an argument.  Which one is correct?

**Miscellaneous.**
* Page 2: "Why is noise-augmented training necessary..." Is it **necessary** for good performance?  It seems to be sufficient, but it's not clear that it is necessary.
* Page 2: "We show that this notion of interference distance captures the trade-offs in randomized smoothing" -- are you asserting that it captures **every possible** trade-off?  Might consider rephrasing.
* Page 2: "...set of samples, typically identically and independently distributed..." -- what is meant by "typically?"  In practice, this does not always hold.  Take MNIST for example.  Is it even possible to determine whether the MNIST digits are IID?
* Page 3: "...as well as the variance of the smoothing distribution $\beta$..." -- $\beta$ is the standard deviation, not the variance.
* Page 5: "...have a very small effect on each other." -- I'm not sure how we can see this from the figure.
* Page 6: The authors should define "simply connected."  Also, the intervals $I$ should be formally defined (with equations).  The definitions of **nice** noise distributions should be moved into the main text.
* Page

---

> ### Author Response · Authors · 2023-03-03
> **Response to Reviewer UVsX: Part 1**
>
> We wholeheartedly appreciate the careful reading of our manuscript, and thank you for finding our paper novel.
>
> > An informal statement of theoretical results can be a useful way of presenting results like this. However, I don't believe that the presentation is effective here.
>
> Thank you for this comment. This style of presenting the results informally first and going into details later was intentional, and effective in our opinion. The contributions are first presented completely colloquially in the introduction, which is then formalized slightly in Section 3, and finally Section 4 presents the results in full technical detail. Moreover, as [Reviewer 9yXc notes](https://openreview.net/forum?id=fvyh6mDWFr&noteId=6peDfiYuZD#:~:text=The%20manuscript%20is%20overall%20well%2Dorganized%20and%20clearly%20written%20%2D%20for%20example%2C%20I%20appreciate%20Section%203%20of%20the%20manuscript%20that%20effectively%20summarizes%20the%20main%20results%20of%20the%20paper.), this makes ``the manuscript well organized, and Sec 3 effectively summarized the main results''. Nevertheless, your point is well taken and we have improved the presentation further by incorporating your suggestions, as follows:
>
> ---
>
> > I would start Section 3 by telling the reader that the results are going to be informally stated.
>
> We had mentioned at the start of Sec 3 (and also in the introduction) that Sec 3 is an informal presentation of the results formalized later in Sec 4. Please see the last line of the first paragraph of Sec 3.
>
> ---
>
> > Then, I would try to make the statements of the results informal. Right now, they read as half-way between formal and informal. It's challenging to interpret the theorems in Section 3 without knowing what the ID is.
>
> We had defined the interference distance informally in the second paragraph of Sec 3, before stating our results. We have improved this description and added the same to the caption of Figure 1.
>
> ---
>
> > In Theorem 3.2, at this level of detail, is this not trivial, in the sense that $G_{\alpha, \beta}$ could simply be taken to be $\Delta_{\alpha, \beta}$? $\Delta_{\alpha, \beta}$ seems to be monotonic in both of these parameters, right?
>
> No, $\Delta_{\alpha, \beta}$ cannot be taken to be $G_{\alpha, \beta}$.  As Th 3.2 states, the latter is only an upper-bound to the former. $\Delta_{\alpha, \beta}$ is a complicated quantity, and the primary focus of the paper is to understand the properties of $\Delta_{\alpha, \beta}$ in different interference distance regimes. In the low interefence distance regime in Th 3.3, $\Delta_{\alpha, \beta}$ is not monotonic in $\alpha$, and a benefit can be obtained by noise-augmented training.
>
> ---
>
> > There is a strange argument at the end of the experiments [...] that since NA is useful, the distribution of data must be in the small ID regime. However, the results in the paper (i.e., Theorems 3.3 and 4.3) seem to say that small ID implies that NA is useful. If we let A denote "the distribution has small ID" and we let B denote "NA is useful", this can be simplified as follows: The theory says that $A \implies B$, whereas the experiments say that $B \implies A$. Clearly, these two are not equivalent. Perhaps the authors can clarify, because this conflict undermines the main results of the paper.
>
> Great observation. Indeed, our choice of language here was imprecise and has led to confusion, as we did not intend to convey that $B \implies A$ from our experiments. We were merely making the observation that the empirical curves for MNIST and CIFAR-10 showed characteristics similar to what one would expect for datasets with small interference distance. We have modified the text to reflect this. We do not believe this undermines any of our results, as the real-data experiments were only meant to serve as high-level empirical support for the conclusions from our theory, which stands on its own regardless.
> However, note that the one dimensional example in Sec 4.2 _does_ define a class of distributions where the interference distance fully determines whether noise-augmentation is beneficial, i.e., $A \Leftrightarrow B$ (see the discussion around Eq. (5) and Eq. (6) in the text).
>
> > Have a clearer take home message that is supported by the experiments
>
> The first take-home message, that is clearly supported by our synthetic experiments in Fig. 4, is that data-augmentation is not always effective in reducing the risk of the classifier after randomized smoothing, and that the interference-distance parameter is able to distinguish between some families of data-distributions where data-augmentation helps randomized smoothing. The second take-home message is, that contrary to intuition, the data-augmentation strength need not be the same as the randomized smoothing strength for best performance of randomized smoothed classifier. This is supported by our theory, as well as observed in our experiments (Fig. 4, 5). We have now stressed upon these messages in the conclusion.

---

> ### Author Response · Authors · 2023-03-03
> **Response to Reviewer UVsX: Part 2**
>
> >Fix the grammatical and notational problems outlined.
>
> Thanks for catching these grammatical problems, we fixed them in the paper, as follows:
>
> > Page 1: "One of the central questions..." -- the authors do not state an actual question here.
>
> Changed to "central object of study".
>
> > Page 2: What is meant by "out of the box?"
>
> Clarified.
>
> > Page 3: "...for a pair of input sample and its label" -- this is awkwardly worded
>
> Fixed wording.
>
> > Page 3: "rendered astray" - this wording doesn't make sense. It would improve readability to use more precise language
>
> Made language precise.
>
> > Page 4: "relaxed the relax this assumption" -- this seems wrong, consider revising
>
> Removed first relax.
>
> > Page 4: "Our first result consists in demonstrating..." -- this is not correct grammar
>
> Fixed grammar.
>
> > Notational problems. The symbol $h$ is used in different ways in some places.
>
> Thanks for catching these errors - $h$ is a conditional distribution, and the notation has been clarified throughout the paper to reflect this.
>
> > In (2), $\Delta$ doesn't have an argument. In Theorem 3.1, it takes $h$ as an argument. Which one is correct?
>
> $\Delta_{\alpha, \beta}(h)$ is the risk of randomized smoothing a classifier obtained from training on noise-augmenting data following the conditional distribution $h$. Thus, it is always dependent on $h$. The notation omitted $h$ at some places in the paper, as we thought it was clear from context -- but we have now modified all occurrences of $\Delta_{\alpha, \beta}$ to contain $h$ as an argument.
>
> ---
>
> > Page 2: "Why is noise-augmented training necessary..." Is it necessary for good performance? It seems to be sufficient, but it's not clear that it is necessary.
>
> Changed "necessary" to "useful".
>
> ---
>
> > Page 2: "We show that this notion of interference distance captures the trade-offs in randomized smoothing" -- are you asserting that it captures every possible trade-off? Might consider rephrasing.
>
> Indeed, we are not asserting that interference distance captures every possible tradeoff. Toned down to "captures some tradeoffs".
>
> ---
>
> > Page 2: "...set of samples, typically identically and independently distributed..." -- what is meant by "typically?" In practice, this does not always hold.
>
> Removed "typically".
>
> > Page 3: $\beta$ is the standard deviation, not the variance.
>
> Fixed.
>
> > Page 5: "...have a very small effect on each other." - I'm not sure how we can see this from the figure.
>
> Pictorially, each positive regions shrink almost uniformly upon smoothing - as if there were no other positive regions. We added this explanation to the text.
>
> > Page 6: The authors should define "simply connected." Also, the intervals
>  should be formally defined (with equations). The definitions of nice noise distributions should be moved into the main text.
>
> "Simply Connected" is defined in the standard topological sense - this definition has now been added to the main text. An explanation for the intervals has been added, and the definition of nice noise distributions has been moved to the main text.
>
> > Add more details to Figures 1 and 2, both of which look great, but are not explained clearly enough.
>
> We have modified the captions of the figures by pointing to detailed explanations of all the panels of the figures in the text. We are happy to iterate further to improve readability if needed.

---

### Review · Reviewer_YbSr · 2023-02-21

**Summary Of Contributions:**

In randomized smoothing, the base classifier is usually trained on images augmented with Gaussian noise of the same magnitude as is applied later during smoothing.  This manuscript aims to understand the purpose of this "noise-augmented training."  The authors introduce a concept called "interference distance" which is intended to capture whether noise-augmented training is helpful or not for the natural accuracy of the smoothed classifier.  On datasets with "low interference distance," like natural image data, noise-augmented training is helpful.  But there do exist datasets with "high interference distance" where noise-augmented training has no benefit and can be harmful.



**Audience:**

Yes

**Broader Impact Concerns:**

no concerns

**Claims And Evidence:**

Yes

**Requested Changes:**

- randomized smoothing was proposed as a certified defense by Lecuyer et al (2019), not Cohen et al (2019) - this paper should be cited on page 1

 - typo in first paragraph of section 4 - you mean the bottom panel

**Strengths And Weaknesses:**

Strengths:
 -  The paper makes a thorough theoretical treatment of a question (whether noise augmented training is necessary for RS) that has not received theoretical attention in the literature.
 - Interestingly, the authors construct situations where noise-augmented training is harmful to the natural accuracy of the smoothed classifier, which is an unintuitive finding.


Weaknesses:
 - As I understand them, Theorems 3.1 and Theorem 3.3 show that _there exist_ distributions with large / small interference distances, such that training on them is not beneficial / beneficial for the natural accuracy of the smoothed classifier.  But this is weaker than showing that interference distance is a complete characterization of when noise-augmented training helps or hurts.

- The specific high-dimensional structure of image data seems different than the synthetic distributions studied in the paper.  In particular, the necessity of Gaussian data augmentation for CIFAR RS strikes me as an OOD accuracy problem - if the base classifier has never seen a noisy image, it won't know how to deal with it at smoothing time.  Could the authors comment?

---

> ### Author Response · Authors · 2023-03-03
> **Response to Reviewer YbSr**
>
> Thank you for finding our work thorough and the studied theoretical question novel. We respond to your comments below:
>
> > As I understand them, Theorems 3.1 and Theorem 3.3 show that there exist distributions with large / small interference distances, such that training on them is not beneficial / beneficial for the natural accuracy of the smoothed classifier. But this is weaker than showing that interference distance is a complete characterization of when noise-augmented training helps or hurts.
>
> Indeed, while our results show that the interference distance is a very useful parameter in determining when noise-augmented training can help randomized smoothing, it is not a complete characterization. In order to make our results as general as possible, we chose not to put additional constraints on the set of data distributions that we study. We believe that a more precise and complete characterization might only be possible by imposing additional distributional assumptions. We have modified the text (P2-Contributions, P12-Future Work) to reinforce this point.
>
> ---
>
> > The specific high-dimensional structure of image data seems different than the synthetic distributions studied in the paper. In particular, the necessity of Gaussian data augmentation for CIFAR RS strikes me as an OOD accuracy problem - if the base classifier has never seen a noisy image, it won't know how to deal with it at smoothing time. Could the authors comment?
>
> Thank you for this comment. Indeed, performing randomized smoothing on a classifier that has not seen "noisy inputs" before is an OOD problem, and our theory studies to what extent this pre-training with noise augmented data is needed. The reviewer's intuition that "the base classifier needs to be trained with noisy images to perform well on noise during randomized smoothing" is precisely what we analyze rigorously in the paper, showing that there exist distributions of the first type (with high interference-distance), where the classifier should not, in fact, be trained with noisy images, and also of the second type (with low interference distance), where the classifier should be trained with noisy images. Hence, a priori, the intuition might be inaccurate for certain families of datasets. Empirical evidence, however, seems to suggest that real image datasets belong to the second type. It is not yet clear why this is true for real-world settings, however, and this remains a topic of future research. Based on this question, we have now added a comment explaining this connection to OOD in the preliminaries.
>
> ---
>
> > Randomized smoothing was proposed as a certified defense by Lecuyer et al (2019), not Cohen et al (2019) - this paper should be cited on page 1.
>
> Thanks, we added this reference.
>
> ---
>
> > Typo in first paragraph of section 4 - you mean the bottom panel.
>
> Indeed, corrected now.

---

### Review · Reviewer_9yXc · 2023-02-25

**Summary Of Contributions:**

The paper theoretically investigates the effect of noise-augmented training in randomized smoothing, which has been a practice to obtain a good smoothed classifier. In particular, it focuses on the change in the optimal benign risk of smoothed classifiers given a (training) noise strength $\alpha$. It proves that noise-augmented training with strength $\alpha$ can be harmful in the benign risk of smoothed classifier for certain data distributions. The paper further characterizes such distributions in terms of “interference distance”, and shows that the favorability of noise-augmented training is partially corresponded to the small interference distance of data distribution. Experimental results demonstrate that one can indeed construct a dataset that noise-augmented training harms the accuracy of smoothed classifiers, while it is also remarked that more realistic data such as MNIST and CIFAR-10 does not suffer from this phenomenon perhaps due to their small enough interference distance.

**Audience:**

Yes

**Broader Impact Concerns:**

None that I am aware of.

**Claims And Evidence:**

Yes

**Requested Changes:**

- A more discussion or (either theoretical or empirical) justification on why real-world data (such as MNIST and CIFAR-10 tested in the paper) tends to get small enough interference distance would help reader to better understand the practical aspects of the theory.
- A discussion compared to [Mohapatra et al., 2021] especially in terms of theoretical novelties would improve clarity.

**Strengths And Weaknesses:**

Strengths

- The manuscript is overall well-organized and clearly written - for example, I appreciate Section 3 of the manuscript that effectively summarizes the main results of the paper.
- The focus of the study, i.e., the effect of noise-augmented training in randomized smoothing, has been sparse in the literature, and the manuscript presents a meaningful theory on this.
- The manuscript also contains empirical supports besides of its interesting theory.

Weaknesses

- To my knowledge, the “shrinking” effect of randomized smoothing (in its decision boundary) was previously explored theoretically by Mohapatra et al. (2021), which is not mentioned in the manuscript. I think the manuscript should include a discussion comparing its theoretical novelty upon the work.
- Although I feel the theory presented in the paper is quite clear, but its practical implication seems relatively weak given the lack of discussion about when the theory holds in the real-world. For example, as empirically shown in the paper, it seems the “no-need-noise-augmentation” theorem does not actually hold in practical scenarios such as MNIST and CIFAR-10. The theory may provide useful insight on the properties of smoothed classifiers, but the week connection to practice can decrease its significance.
- (minor) Some lines are somewhat unclear to me:
    - p.3: “this implies a minimal strength for the randomized smoothing parameter, say $\beta^*$” - why one requires a minimal parameter given $\epsilon^*$? How does $\beta^*$ actually defined?
    - Theorem 3.3: What denotes $H$ in $\zeta_{H}$, perhaps a typo?
    - Figure 4: “The line is dashed if … for any $\alpha > 0$, …” - shouldn’t be that “there exist $\alpha$ …” instead?

[Mohapatra et al., 2021] Hidden Cost of Randomized Smoothing, AISTATS 2021.

---

> ### Author Response · Authors · 2023-03-04
> **Response to Reviewer 9yXc**
>
> We thank you for finding our theory interesting, and our manuscript well-written. We respond to your comments below.
>
> > To my knowledge, the "shrinking" effect of randomized smoothing (in its decision boundary) was previously explored theoretically by Mohapatra et al. (2021), which is not mentioned in the manuscript. I think the manuscript should include a discussion comparing its theoretical novelty upon the work. A discussion compared to [Mohapatra et al., 2021] especially in terms of theoretical novelties would improve clarity.
>
> Thanks for the reference - indeed (Mohapatra et al. (2021)) is related to our work, and we have now commented on it in the related work section. To summarize, (Mohapatra et al. (2021)) derive their theoretical results by demonstrating that randomized smoothing leads to shrinkage of the decision boundaries of one of the classes for simple data distributions. Importantly, their definition of "shrinkage" is that the bounding sphere (or cone, for "semi-bounded" regions) of the decision boundary becomes smaller. This is different from our analysis, as a shrunk decision region $R_\sigma$ might not be a subset of the original region $R$ under this definition. On the other hand, our universal result 4.2 holds in much more generality for arbitrary (bounded) data-distributions, and our existence results 4.1, 4.3 construct specific synthetic datasets. We work with the risk directly, and shrinkage shows up in some parts of our analysis on the path to deriving bounds on the risk of the randomized smoothed classifier. Additionally, and perhaps more importantly, our proof techniques handle both data-augmentation of strength $\alpha$ and randomized smoothing of strength $\beta$ simultaneously, allowing us to discover cases where data-augmentation helps randomized smoothing (in other words, showing cases where the combined effect might not be a shrinkage of the decision boundaries).
>
> ---
>
> > Although I feel the theory presented in the paper is quite clear, but its practical implication seems relatively weak given the lack of discussion about when the theory holds in the real-world. A more discussion or (either theoretical or empirical) justification on why real-world data (such as MNIST and CIFAR-10 tested in the paper) tends to get small enough interference distance would help reader to better understand the practical aspects of the theory.
>
> Thank you for this great comment. First, note that the main point of our "no-need for noise-augementation" result is to formalize the observation that there exist distributions where noise-augmentation is not needed for randomized smoothing to perform well. We believe this is interesting since, if one were to judge only from known results (and empirical evidence on real datasets), one would be inclined to believe that noise augmentation is always needed.
>
> On the other hand, we do not have a formal understanding of why real-world datasets like MNIST and CIFAR-10 seem to have to low-interference distance, but our conjecture is that the observed behavior on MNIST and CIFAR-10 is a function of both the datasets and the structure of the trained classifier. For example, it is possible that high dimensional neural networks trained without sufficient regularization tend to create decision regions that "wrap" tightly around clusters of data-points. In our setting, this translates to the disjoint positive regions being large in number, and close to each other, with very little space between each other in the ambient space. While this picture is only approximate, this could lead to a partition of the space with low interference distance.
>
> ---
>
> > (minor) p.3: “this implies a minimal strength for the randomized smoothing parameter, say $\beta^*$ - why one requires a minimal parameter given $\epsilon^*$? How is actually $\beta^*$ defined?
>
> We consider a scenario where a minimum level of robustness is warranted; i.e., we would like to have a certain minimum certified radius $\epsilon^*$,  guaranteeing robustness against any $\ell_2$ perturbation of size $\epsilon^*$. As the certificate arising from randomized smoothing is given by $\beta \Phi^{-1}(s)$, we should have $\beta \Phi^{-1}(s) \geq \epsilon^* \implies \beta \geq (1/\Phi^{-1}(s))\epsilon^*$. In other words, to provide this level of certified robustness, we need a smoothing strength of at least $\beta^* \overset{{\rm def}}{=} (1/\Phi^{-1}(s)) \epsilon^*$. We have added this explanation to the manuscript.
>
> ---
>
> > (minor) Theorem 3.3: What denotes $H$ in $\zeta_H$? perhaps a typo?
>
> This was indeed a typo -- thanks for noticing, fixed now.
>
> ---
>
> > (minor) Figure 4: “The line is dashed if … for any $\alpha>0$, …” - shouldn’t be that “there exist $\alpha$ …” instead?
>
> Indeed, this is clearer, thanks for the suggestion. It is modified now.

---

### Decision · Action_Editors · 2023-04-10

**Recommendation:** Accept as is

**Comment:**

The paper theoretically investigates the effect of noise-augmented training in randomized smoothing, which has been a practice to obtain a good smoothed classifier. The authors introduce a concept called "interference distance" which is intended to capture whether noise-augmented training is helpful or not for the natural accuracy of the smoothed classifier. Experimental results demonstrate that one can indeed construct a dataset that noise-augmented training harms the accuracy of smoothed classifiers, while it is also remarked that more realistic data such as MNIST and CIFAR-10 does not suffer from this phenomenon.

All reviewers suggests "Accept" or "Leaning Accept" as the authors' claims are technically correct and would be of interest for some TMLR audience. AC agrees with this and also recommend acceptance. However, AE and some reviewers think that the paper would be much stronger if the experimental evidence was more compelling to practitioners.

**Audience:**

Noise-augmented training is one of widely used techniques for obtaining a robust classifier under randomized smoothing. Hence, the authors' unintuitive claims and findings should be of interest for the TMLR's audience working on ML safety.

**Claims And Evidence:**

The claims made in the submission are technically correct with formal proofs.